# Part-Aware Bottom-Up Group Reasoning for Fine-Grained Social Interaction Detection

**Dongkeun Kim**    **Minsu Cho**    **Suha Kwak**
Pohang University of Science and Technology (POSTECH)
{kdk1563, mscho, suha.kwak}@postech.ac.kr

## Abstract

Social interactions often emerge from subtle, fine-grained cues such as facial expressions, gaze, and gestures. However, existing methods for social interaction detection overlook such nuanced cues and primarily rely on holistic representations of individuals. Moreover, they directly detect social groups without explicitly modeling the underlying interactions between individuals. These drawbacks limit their ability to capture localized social signals and introduce ambiguity when group configurations should be inferred from social interactions grounded in nuanced cues. In this work, we propose a part-aware bottom-up group reasoning framework for fine-grained social interaction detection. The proposed method infers social groups and their interactions using body part features and their interpersonal relations. Our model first detects individuals and enhances their features using part-aware cues, and then infers group configuration by associating individuals via similarity-based reasoning, which considers not only spatial relations but also subtle social cues that signal interactions, leading to more accurate group inference. Experiments on the NVI dataset demonstrate that our method outperforms prior methods, achieving the new state of the art, while additional results on the Café dataset further validate its generalizability to group activity understanding.

## 1 Introduction

Understanding the intentions and behaviors of others is a fundamental aspect of human perception and social intelligence. In our daily lives, we rely not only on what people say, but also on how they appear, move, and behave to make sense of others. Understanding such interactions enables broad applications, including surveillance, human-robot interaction, and sports analysis. Social interaction understanding encompasses a wide range of tasks, including group activity recognition [13, 19, 23, 30, 54, 56], pedestrian trajectory prediction [1, 2, 37, 52] and group activity detection [10, 11, 24, 45]. While these tasks have advanced the ability to model social interactions, most existing work primarily focuses on group behaviors inferred from coarse visual cues such as appearances, actions, or geometric configurations of group members. Consequently, they often overlook fine-grained social interactions, such as identifying whether two people are smiling at each other, engaging in mutual gaze, or performing gestures like pointing, that are essential for nuanced social perception. The ability to understand such fine-grained social interactions is crucial for inferring intent, emotion, and social relationships, particularly when verbal communication is limited or absent.

Recently, a new task of detecting multi-person interactions based on such fine-grained, nuanced, and ambiguous social cues has been introduced along with a dedicated dataset, NVI [51]. The task, named *nonverbal interaction detection* (NVI-DET), is formulated to detect each individual, identify the social group they belong to, and classify their fine-grained social interaction, encapsulated as a triplet `<individual, group, interaction>`. This formulation draws inspiration from human–object interaction (HOI) detection [31, 44, 58], but differs in that it targets human–human social interactions, including facial expression, gesture, posture, gaze, and touch, that are inherently more nuanced and

39th Conference on Neural Information Processing Systems (NeurIPS 2025).

relational. The task poses unique challenges, requiring both accurate localization of individuals and the interpretation of subtle cues that define social groups and their interactions.

Previous approaches [31, 51] attempt to solve this task by utilizing transformer [5, 47] and hyper-graph [61] to capture high-order interactions among individuals and social groups. While these approaches are effective in capturing social interactions among individuals and groups, they have several drawbacks as follows. First, these methods directly detect social groups without explicitly modeling the underlying person-to-person relations. This design overlooks a fundamental principle of social interaction detection: both social interactions and social group composition should emerge from individual behaviors and interactions between individuals. Predicting a group without accounting for its members introduces ambiguity and uncertainty, especially when subtle interactions like gaze occur between individuals who are spatially distant. Second, most existing methods embed each person as a holistic representation, neglecting body parts information that are essential for recognizing fine-grained social interactions. As a result, they struggle to distinguish interactions seemingly alike but holding different semantics stemming from subtle visual cues, such as 'mutual gaze' versus 'gaze following' or 'wave' versus 'point,' that require fine-grained social reasoning.

To address these limitations, we propose a part-aware bottom-up group reasoning model for fine-grained social interaction detection. The core idea of our model is to infer group composition and interactions by reasoning from fine-grained body part cues to relations between individuals. Instead of directly detecting both individuals and social groups, we first detect individuals and construct group representations by aggregating individual-level information based on learned similarities, allowing social groups to naturally emerge from the interactions between individuals. Moreover, our model enriches individual embeddings by incorporating part-aware features, which are learned under the guidance of human pose estimation as privileged information [34] to attend specifically to different body parts such as face, arms, and legs. These representations encode localized body part cues such as facial expression, hand gesture, and posture, that are crucial for recognizing fine-grained social interactions. The proposed group reasoning framework reflects the compositional nature of social interactions and enables the model to detect fine-grained social interactions with improved performance.

We evaluated the proposed method on the NVI [51] and Café [24], where it demonstrated substantial improvements over existing methods. In summary, our contribution is three-fold as follows:

- We introduce a part-aware representation learning that leverages pose-guided supervision as privileged information to capture fine-grained social cues for improving interaction recognition.

- We propose a bottom-up group reasoning framework that infers social groups based on part-aware individual representations, rather than directly detect social groups. This design ensures that group composition naturally emerges from individuals.

- The proposed method outperforms existing approaches on NVI and Café, demonstrating the effectiveness of incorporating body part representations and bottom-up group reasoning for fine-grained social interaction detection.

## 2   Related work

### 2.1   Social interaction understanding

**Fine-grained social interaction recognition.** Fine-grained social interaction recognition plays a central role in interpreting intent, behavior, and social dynamics [3]. Recent advances in computer vision have explored fine-grained social cues that are subtle yet essential for human communication, including gaze, facial expressions, and gestures. Gaze analysis has been a representative line of work in fine-grained social interaction understanding, which aims to localize where individuals direct their attention [6, 7, 15, 20, 43]. Specifically, MTGS [15] introduces a unified dataset for multi-person gaze following and social gaze prediction, while Tafasca *et al.* [43] extend the gaze following task by jointly predicting both the location and semantic label of the gaze target. Facial expression [22, 50, 60], gesture [14, 38, 59] and posture [9] analysis form another line of research that targets specific types of fine-grained social interactions. While these methods have shown success, they are typically developed on specialized datasets that focus on specific signals.

**Group activity understanding.** Group activity recognition (GAR) has long been studied as a representative group activity understanding task, aiming to classify collective activity exhibited by groups of people. Typical GAR approaches model spatio-temporal relations between individuals utilizing graph neural networks [54, 56, 57] or transformers [13, 23, 30]. However, these methods are built on the assumption of a single-group setting, which restricts their applicability to more complex, real-world scenarios. Beyond recognizing a single group activity, some work tackles multi-group scenarios in the form of social activity recognition [11, 45, 56] or group activity detection [24], which aims to localize multiple groups and classify the activity performed by each group. While group activity detection shares similarity with NVI-DET in that it localizes multiple social groups in a scene, most of these methods treat individuals as holistic units and focus on activity-level classification, overlooking fine-grained social cues.

**Nonverbal interaction detection.** Fine-grained social signals are a rich yet under-explored cue for understanding social interactions [48]. Wei *et al.* [51] introduce NVI-DET task, which seeks to detect fine-grained social interactions by identifying triplets ⟨individual,group,interaction⟩ in an image. This formulation enables unified reasoning over individual and social interaction through visually grounded nonverbal cues. The accompanying NVI dataset includes annotations for five interaction categories: facial expression, gesture, posture, gaze, and touch. The proposed model, NVI-DEHR, leverages hypergraphs [61] to model high-order individual-to-individual and group-to-group relations, improving its ability to recognize fine-grained social interactions. However, NVI-DEHR detects social groups directly without explicitly modeling inter-person relations—a limitation that becomes particularly problematic when detecting interactions like gaze, which often occur between spatially distant individuals. Moreover, it relies on holistic person representations, thereby overlooking body part-level cues that are essential for distinguishing visually similar but semantically distinct interactions (*e.g.*, *wave vs. point*, or *mutual gaze vs. gaze following*). Unlike this approach, our method adopts a hierarchical reasoning strategy that first extracts part-aware individual representations and then infers social groups and their interactions based on inter-personal relations guided by fine-grained body part cues.

## 2.2 Human-object interaction detection

Human-object interaction detection (HOI-DET) [4, 12, 16, 21, 25, 29, 31, 36, 44, 58], which predicts ⟨human,object,interaction⟩ triplets, is closely related to NVI-DET. Indeed, HOI-DET and NVI-DET share structural similarities in their problem formulation and modeling strategies, such as the use of set-based prediction and relational reasoning. However, HOI-DET primarily considers pairwise relations between a single human and an object whereas NVI-DET requires group-aware reasoning among multiple humans. Meanwhile, recent HOI-DET methods explore fine-grained reasoning to improve interaction understanding. For instance, Wan *et al.* [49] utilize an off-the-shelf pose estimator to zoom into relevant body parts. Wu *et al.* [53] incorporate pose cues to better capture spatial configuration between humans and objects, highlighting the benefit of human prior in interaction modeling. Lei *et al.* [28] leverages large language model (LLM) to reason over body-part contexts, enabling the model to associate specific interaction types with relevant body regions. However, unlike these methods, our model leverages human pose only for training, as privileged information [34]. This design allows the model to benefit from fine-grained supervision while maintaining efficient inference without requiring additional inputs.

## 3 Proposed method

We propose a part-aware bottom-up group reasoning framework for fine-grained social interaction detection. The core idea of our framework lies in the bottom-up inference of group configurations and interactions, by leveraging pose-guided part-aware representations and modeling their interpersonal relations. This section describes the problem formulation of NVI-DET (Sec. 3.1), overall architecture of the proposed model (Sec. 3.2), training objectives (Sec. 3.3), and inference procedure (Sec. 3.4).

## 3.1 Problem formulation

The task of fine-grained social interaction detection [51] aims to recognize interactions of each individual with their respective social groups. Given an input image $\mathbf{X} \in \mathbb{R}^{H_0 \times W_0 \times 3}$, the goal is to predict a set of triplets $\mathcal{Y} = \{(\mathbf{b}_i, \mathbf{g}_i, \mathbf{c}_i) | \mathbf{b}_i \in \mathbb{R}^4, \mathbf{g}_i \in \mathbb{R}^4, \mathbf{c}_i \in \mathbb{R}^{N_C}, i \in [1, N]\}$, where each

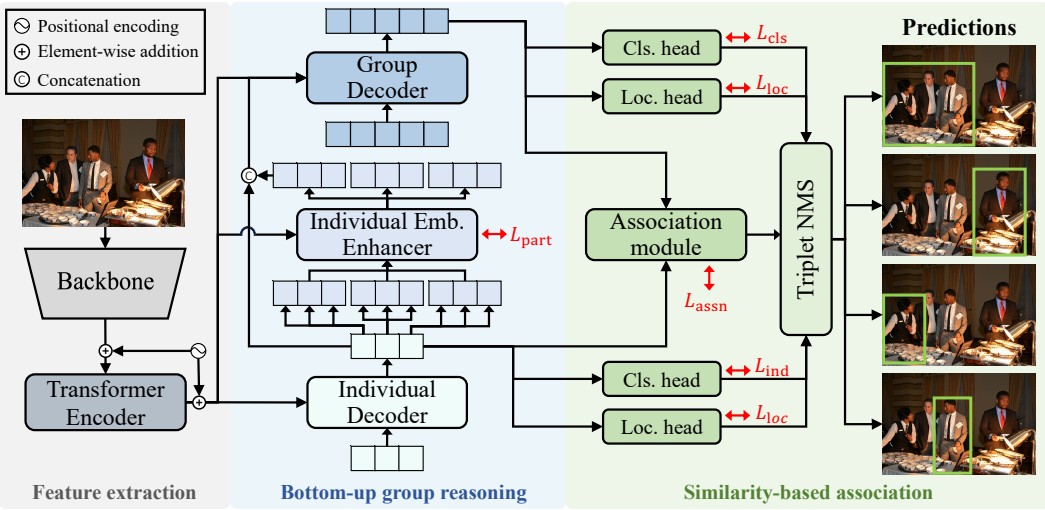

Figure 1: Overall architecture of the proposed framework. Given an input image, our model extracts the visual features using the backbone and the transformer encoder. It then detects individuals and derives part-aware individual features through the individual embedding enhancer. Group queries attend to both the encoded feature maps and part-aware individual embeddings to infer social groups and interaction labels. Finally, triplets are obtained through the association module and NMS.

triplet $(\mathbf{b}_i, \mathbf{g}_i, \mathbf{c}_i)$ consists of an individual bounding box $\mathbf{b}_i$, the corresponding group bounding box $\mathbf{g}_i$, and interaction probabilities $\mathbf{c}_i$ over $N_C$ predefined interaction classes. The interaction probabilities indicate the presence of each interaction for each detected group. Notably, this formulation allows a single individual to participate in multiple concurrent social groups and interactions, similar to HOI-DET [16].

## 3.2 Model architecture

We introduce a part-aware bottom-up framework that detects social interactions by progressively capturing subtle body part cues and leveraging them to infer social groups and their fine-grained interactions. Fig. 1 illustrates the overall pipeline of our model. Unlike prior methods that directly regress group bounding boxes and rely solely on holistic human features, our method first detects individuals and enriches their features with body part cues to ground fine-grained social cues, and then infers group association and interactions through part-aware reasoning.

**Feature extractor.** Given an input image $\mathbf{X} \in \mathbb{R}^{H_0 \times W_0 \times 3}$, we extract a feature map using a ResNet [17] backbone pretrained on ImageNet [8], followed by a standard transformer encoder [5, 47] composed of $L$ layers of multi-head self-attention and feed-forward network (FFN). To align the dimension of the feature map with the dimension of the transformer encoder, a linear projection is applied before feeding the features into the encoder. The resulting encoded feature map $\mathbf{F} \in \mathbb{R}^{H \times W \times D}$ enriches the visual features for subsequent reasoning modules.

**Individual decoder.** Unlike prior work [51], which predicts both individual and group bounding boxes simultaneously, our individual decoder is dedicated solely to detecting individuals in the image. We argue that directly predicting a group bounding box without modeling the interactions among individuals is often ambiguous and counterintuitive. For instance, social groups engaged in mutual gaze or pointing may be spatially distant, and naïvely predicting a bounding box around them can inadvertently include unrelated people. To address this, we adopt a sequential approach: we first detect individuals and subsequently infer groups based on their interactions. The individual decoder adopts the standard transformer decoder architecture of DETR [5]. It employs a set of $N_I$ learnable queries $\mathbf{Q}_I \in \mathbb{R}^{N_I \times D}$, which attend to the encoded image features to produce individual embeddings $\mathbf{E}_I \in \mathbb{R}^{N_I \times D}$. Each output embedding is then passed through a feed-forward network (FFN) to predict the corresponding individual bounding box coordinates $\mathbf{b} \in \mathbb{R}^{N_I \times 4}$.

**Individual embedding enhancer.** To address the fine-grained social interaction detection, which depends on subtle social cues from specific body parts such as the face, eyes, and hands, we introduce the individual embedding enhancer, which incorporates localized, body part-aware features into each

detected individual. Instead of relying solely on holistic representation, which can obscure subtle cues essential for distinguishing fine-grained social interactions, we decompose their appearance into distinct body parts such as face, arms, hands, and legs. Given individual embedding $\mathbf{E}_I$, we generate $P$ part-specific queries for each individual using a set of learnable linear projections:

$$\mathbf{Q}_P = \mathbf{E}_I \cdot [\mathbf{W}_1, \mathbf{W}_2, \ldots, \mathbf{W}_p, \ldots, \mathbf{W}_P] \in \mathbb{R}^{N_I \times P \times D}, \tag{1}$$

where $\mathbf{W}_p \in \mathbb{R}^{D \times D}$ is a learnable weight matrix for the $p$-th body part. The enhancer refines the part queries $\mathbf{Q}_P$ via self-attention across part queries and cross-attention with the encoded feature map $\mathbf{F}$, producing the part embedding $\mathbf{E}_P \in \mathbb{R}^{N_I \times P \times D}$. We then concatenate the part embeddings $\mathbf{E}_P$ with the corresponding individual embeddings $\mathbf{E}_I$ to obtain part-aware individual features:

$$\mathbf{E}_A = [\mathbf{E}_I, \mathbf{E}_P^1, \ldots, \mathbf{E}_P^p, \ldots, \mathbf{E}_P^P] \cdot \mathbf{W}_{\text{fuse}} \in \mathbb{R}^{N_I \times D}, \tag{2}$$

where $\mathbf{E}_P^p$ denotes the $p$-th body part embedding and $\mathbf{W}_{\text{fuse}} \in \mathbb{R}^{(P+1)D \times D}$ is a learnable projection matrix. The resulting part-aware individual embedding $\mathbf{E}_A$ captures both holistic features of the individual (*e.g.*, appearance, position) and localized body cues (*e.g.*, facial expression, gaze, and gestures), which are critical for inferring group association and fine-grained social interaction classes.

**Group decoder.** To address the difficulty of directly regressing group bounding boxes without modeling interpersonal interactions, the group decoder performs group association and interaction recognition by adopting part-aware bottom-up group reasoning approach. Unlike a standard decoder that only attends to visual features, our group decoder leverages a richer context by attending to both the encoded image features $\mathbf{F}$ and the part-aware individual embeddings $\mathbf{E}_A$, enabling the model to extract group-level features by aggregating information from relevant individuals including their body parts and the feature map. Specifically, the group decoder utilizes a set of $N_G$ learnable group queries $\mathbf{Q}_G \in \mathbb{R}^{N_G \times D}$ to produce group embeddings $\mathbf{E}_G$, each of which is further decoded into two outputs: (1) predicted group bounding box coordinates $\mathbf{g} \in \mathbb{R}^{N_G \times 4}$ and (2) multi-label classification scores over the predefined fine-grained social interaction classes $\mathbf{c} \in \mathbb{R}^{N_G \times N_C}$.

**Similarity-based association.** To associate each detected individual with its corresponding social group, we adopt a similarity-based association approach. The individual embedding $\mathbf{E}_I$ and the group embedding $\mathbf{E}_G$ are separately projected and dot-producted to yield a similarity matrix:

$$\mathbf{S} = \text{MLP}(\mathbf{E}_G) \cdot \text{MLP}(\mathbf{E}_I)^T \in \mathbb{R}^{N_G \times N_I}. \tag{3}$$

This matrix represents the affinity between each group and individual, enabling the model to identify the representative individual for each social groups based on learned similarities. For each predicted group, we select the individual with the highest similarity as the representative individual. Unlike prior work that directly predicts group bounding boxes as spatial proposals using group queries [51], our method infers group configuration in a bottom-up manner driven by fine-grained social cues and interpersonal relations.

## 3.3 Training objective

Our model is trained with five losses: $\mathcal{L}_{\text{ind}}$ for individual objectness, $\mathcal{L}_{\text{cls}}$ for multi-label interaction classification, $\mathcal{L}_{\text{loc}}$ for individual and group bounding box localization, $\mathcal{L}_{\text{part}}$ for body part supervision, and $\mathcal{L}_{\text{assn}}$ for group association. All losses are computed between predictions and ground-truth instances matched via the Hungarian algorithm [27].

**Standard NVI losses.** Three among the five losses, namely $\mathcal{L}_{\text{ind}}$, $\mathcal{L}_{\text{cls}}$, and $\mathcal{L}_{\text{loc}}$, are adopted from the previous work on NVI-DET [51]. $\mathcal{L}_{\text{ind}}$ and $\mathcal{L}_{\text{cls}}$ employ focal loss [32] and asymmetric loss (ASL) [42], respectively, to address the long-tailed and imbalanced nature of the labels. The localization loss $\mathcal{L}_{\text{loc}}$ is computed as a weighted sum of $\ell_1$ and generalized IoU loss [41].

**Part loss.** To supervise each part query to focus on distinct body regions, we adopt pose-guided pseudo-supervision that guides the attention toward their corresponding areas. Fig. 2 illustrates the overall pipeline of this pose-guided binary mask generation. We used ViTPose [55] to extract keypoints for each detected individual, other pose estimation models could also be used though; the pose estimation model is used only for training as privileged information [34, 46], and thus imposes no additional space-time complexity in testing. To generate supervision masks from keypoints, we define a square window centered at each keypoint location. The size of the window is set proportional to the size of the corresponding individual box, calculated as: $s_i = \alpha \cdot \max(w_i, h_i)$, where $(w_i, h_i)$ are the width and height of the $i$-th individual's bounding box. The binary mask $M_i^p$ is defined over the feature map such that:

$$M_i^p = \begin{cases} 1 & \text{if } |u - x_i^p| \leq \frac{s_i}{2} \text{ and } |v - y_i^p| \leq \frac{s_i}{2}, \\ 0 & \text{otherwise,} \end{cases} \quad (4)$$

where $(x_i^p, y_i^p)$ is the keypoint location for the $p$-th part of the $i$-th individual, and $(u, v)$ is a spatial coordinate on the feature map. A mean squared error (MSE) loss is then computed between each part query's attention map $A_i^p$ and its corresponding mask $M_i^p$:

$$\mathcal{L}_{\text{part}} = \frac{1}{N_I P} \sum_{i=1}^{N_I} \sum_{p=1}^{P} \|A_i^p - M_i^p\|_2^2, \quad (5)$$

where $A_i^p$ is the attention map of the $p$-th part query for the $i$-th instance, and $M_i^p$ is the corresponding pseudo ground-truth mask. This supervision promotes localized attention over the human body, which is particularly beneficial for recognizing interaction types grounded in specific body parts, including face, shoulders, elbows, wrists, hips, knees, and ankles.

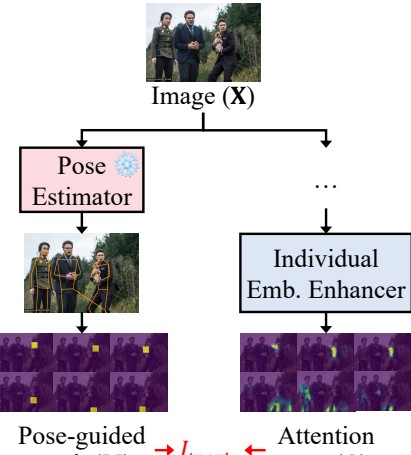

Figure 2: Pose-guided binary mask generation for part supervision.

**Association loss.** To address the ambiguity of directly regressing group bounding boxes, we associate individuals into a group based on their similarity scores. We use a binary cross-entropy (BCE) loss between the predicted and ground-truth similarity scores, computed only over matched groups and matched individuals. Let $\sigma(i)$ be the predicted group matched to the $i$-th ground-truth group, and let $\tau(j)$ be the predicted individual matched to the $j$-th ground-truth individual. The association loss is defined as:

$$\mathcal{L}_{\text{assn}} = -\frac{1}{|\mathcal{I}||\mathcal{J}|} \sum_{i \in \mathcal{I}} \sum_{j \in \mathcal{J}} \left( \mathbf{a}_i(j) \log \mathbf{S}_{\sigma(i)}(\tau(j)) + (1 - \mathbf{a}_i(j)) \log(1 - \mathbf{S}_{\sigma(i)}(\tau(j))) \right), \quad (6)$$

where $\mathbf{a}_i(j) = 1$ if the $j$-th individual belongs to the $i$-th group, $\mathcal{I}$ and $\mathcal{J}$ denote the set of groups and the set of individuals, respectively. This loss encourages the group embedding to attend to the individuals that belong to the corresponding group in the group decoder, by increasing the similarity between their embeddings.

**Total loss.** Our model is trained with five losses simultaneously in an end-to-end manner. Specifically, the total training objective is a linear combination of the five losses as follows:

$$\mathcal{L} = \lambda_i \mathcal{L}_{\text{ind}} + \lambda_c \mathcal{L}_{\text{cls}} + \lambda_l \mathcal{L}_{\text{loc}} + \lambda_p \mathcal{L}_{\text{part}} + \lambda_a \mathcal{L}_{\text{assn}}. \quad (7)$$

### 3.4 Inference

At inference time, we associate the predicted individual and group embeddings to generate final predictions in the form of <individual, group, interaction> triplets. To associate individuals with predicted group embeddings, we use the similarity matrix $\mathbf{S} \in \mathbb{R}^{N_G \times N_I}$ computed between their embeddings. For each predicted group $i$, we construct a triplet as: $\langle \mathbf{b}_{j*}, \mathbf{g}_i, \mathbf{c}_i \rangle$, where $\mathbf{b}_{j*}$ denotes the predicted bounding box coordinates of the individual with the highest similarity $j^* = \arg\max_j \mathbf{S}_i(j)$, $\mathbf{g}_i$ represents the predicted group box coordinates, and $\mathbf{c}_i$ is the predicted interaction logit. Finally, we apply triplet NMS to remove redundant predictions: triplets are suppressed if their individual and group bounding boxes, as well as predicted interaction labels, significantly overlap with those of a triplet with higher interaction scores.

## 4 Experiments

### 4.1 Experimental settings

**Dataset.** To verify the proposed method across diverse social scenarios, we evaluated on two benchmarks: NVI [51] and Café [24]. NVI contains 13,711 images, with 9,634 for training, 1,418 for validation, and 2,659 for test. It defines 22 atomic-level interaction classes grouped into 5 categories:

Table 1: Comparison with the state-of-the-art methods on the NVI dataset.

| Method | val | | | | test | | | |
|---|---|---|---|---|---|---|---|---|
| | mR@25 | mR@50 | mR@100 | AR | mR@25 | mR@50 | mR@100 | AR |
| *m*-QPIC [44] | 56.89 | 69.52 | 78.36 | 68.26 | 59.44 | 71.46 | 80.07 | 70.32 |
| *m*-CDN [58] | 55.57 | 71.06 | 78.81 | 68.48 | 59.01 | 72.94 | 82.61 | 71.52 |
| *m*-GEN-VLKT [31] | 50.59 | 70.87 | 80.08 | 67.18 | 56.68 | 74.32 | 84.18 | 71.72 |
| NVI-DEHR [51] | 54.85 | 73.42 | 85.33 | 71.20 | 59.46 | 76.01 | 88.52 | 74.67 |
| **Ours** | **59.43** | **76.62** | **87.43** | **74.49** | **63.59** | **80.62** | **91.34** | **78.52** |

Table 2: Comparison with the state-of-the-art methods on the Café dataset under the detection-based setting. Scores are from [24].

| Method | Split by view | | | Split by place | | |
|---|---|---|---|---|---|---|
| | Group $mAP_{1.0}$ | Group $mAP_{0.5}$ | Outlier mIoU | Group $mAP_{1.0}$ | Group $mAP_{0.5}$ | Outlier mIoU |
| Joint [10] | 9.14 | 31.83 | 42.93 | 6.08 | 18.43 | 2.83 |
| JRDB-base [11] | 12.63 | 35.53 | 31.85 | 8.15 | 22.68 | 33.03 |
| HGC [45] | 6.77 | 31.08 | 57.65 | 4.27 | 24.97 | 57.70 |
| Café-base [24] | 14.36 | 37.52 | 63.70 | 8.29 | 28.72 | 59.60 |
| **Ours** | **18.23** | **46.88** | **67.62** | **10.65** | **39.03** | **63.60** |

*facial expression*, *gesture*, *posture*, *gaze*, and *touch*. These include both 16 individual-level and 6 group-level interactions, enabling the analysis of both fine-grained and group-level reasoning for fine-grained social interaction detection. Café is a large-scale multi-view, multi-person video benchmark for group activity detection. Each clip contains multiple co-occurring groups performing distinct activities, allowing us to evaluate whether the proposed part-aware reasoning can generalize from fine-grained social interactions to broader group activity understanding.

**Evaluation metrics.** Following NVI-DET protocol, we report mean Recall@$K$ ($K = \{25, 50, 100\}$), and their average (AR). Each recall is averaged over three IoU thresholds: 0.25, 0.5, and 0.75. For Café, we use Group mAP at Group IoU thresholds of 0.5 and 1.0, and Outlier mIoU, which jointly capture accuracy in detecting multiple simultaneous group activities.

**Hyperparameters.** Our model is initialized with the pretrained DETR ResNet-50. The feature dimension $C$ and the transformer dimension $D$ are set to 2048 and 256, respectively. The encoder consists of 6 layers with 8 attention heads, while the individual decoder, individual embedding enhancer, and group decoder comprise 3 layers with 8 attention heads. The number of individual queries, group queries, and parts are 24, 32, and 13, respectively. The NMS threshold is set to 0.5.

**Training.** We train our model for 90 epochs using the AdamW optimizer [35] with $\beta_1 = 0.9$, $\beta_2 = 0.999$, and $\epsilon = 1e-8$. The learning rate is set to $1e-4$ initially and decayed to $1e-5$ after 60 epochs. We use a mini-batch size of 4. Loss coefficients are set to $\lambda_i = 1.0$, $\lambda_c = 2.0$, $\lambda_l = 1.0$, $\lambda_{\ell_1} = 2.5$, $\lambda_{GIoU} = 1.0$, $\lambda_p = 10.0$, and $\lambda_a = 5.0$. For part supervision, we use ViTPose [55] to extract 13 keypoints per person, excluding four facial keypoints (`left-eye`, `right-eye`, `left-ear`, `right-ear`) to avoid spatial overlap in pseudo-supervision masks. The window size proportion $\alpha$ is set to 0.2 relative to the size of the individual box.

## 4.2 Quantitative analysis

We compare our method against three HOI-DET baselines, *m*-QPIC [44], *m*-CDN [58], and *m*-GEN-VLKT [31], adapted to the NVI-DET setting, as well as the current state-of-the-art NVI-DET model, NVI-DEHR [51]. Table 1 presents performance comparisons on the NVI dataset [51], where our model outperforms all the others by substantial margins on both validation set and test set. Specifically, our method outperforms NVI-DEHR by 3.29 in AR on the validation set and 3.85 in AR on the test set. Notably, the gains are even more pronounced in mR@25, with improvements of 4.58 and 4.13 on the validation set and test set, respectively. It demonstrates the efficacy of our part-aware bottom-up group reasoning framework in both detecting social groups and inferring fine-grained social interactions.

Table 3: Comparison with MLLMs.

| Method | mR@25 | mR@50 | mR@100 | AR |
|---|---|---|---|---|
| **Ours** | **63.59** | **80.62** | **91.34** | **78.52** |
| LLaVA [33] | 21.09 | 36.75 | 53.59 | 37.14 |
| LLaVA-LoRA [18] | 17.40 | 32.12 | 51.93 | 33.81 |

Table 4: Impact of pose supervision.

| Setting | mR@25 | mR@50 | mR@100 | AR |
|---|---|---|---|---|
| **Ours** | **59.43** | 76.62 | **87.43** | **74.49** |
| VLM | 55.18 | **76.94** | 86.96 | 73.02 |

Table 5: Impact of the proposed module.

| Setting | mR@25 | mR@50 | mR@100 | AR |
|---|---|---|---|---|
| **Ours** | **59.43** | **76.62** | 87.43 | **74.49** |
| w/o enhancer | 55.20 | 73.25 | **88.05** | 72.17 |
| w/o sim-assn | 55.95 | 74.95 | 87.36 | 72.75 |
| w/o both | 56.29 | 70.52 | 85.77 | 70.86 |

Table 6: Impact of the loss functions.

| Setting | mR@25 | mR@50 | mR@100 | AR |
|---|---|---|---|---|
| **Ours** | **59.43** | 76.62 | 87.43 | **74.49** |
| w/o $\mathcal{L}_{assn}$ | 30.38 | 47.20 | 64.49 | 48.32 |
| w/o $\mathcal{L}_{loc}$ | 49.56 | 71.73 | 83.59 | 68.29 |
| w/o $\mathcal{L}_{part}$ | 54.32 | **78.80** | **87.60** | 73.58 |

Table 7: Impact of the number of parts.

| $P$ | mR@25 | mR@50 | mR@100 | AR |
|---|---|---|---|---|
| 5 | 57.01 | 75.29 | 87.01 | 73.11 |
| 9 | 54.59 | **76.69** | 87.88 | 73.05 |
| **13 (Ours)** | **59.43** | 76.62 | 87.43 | **74.49** |
| 17 (All) | 54.87 | 76.30 | **88.14** | 73.10 |

Table 8: Impact of the number of queries.

| $N_I$ | $N_G$ | mR@25 | mR@50 | mR@100 | AR |
|---|---|---|---|---|---|
| 24 | 24 | 58.94 | 76.57 | 86.94 | 74.15 |
| **24** | **32** | **59.43** | **76.62** | 87.43 | **74.49** |
| 32 | 24 | 58.06 | 76.59 | **88.01** | 74.22 |
| 32 | 32 | 56.56 | 75.81 | 86.95 | 73.11 |

Table 2 shows experiments on Café [24], a recent and challenging benchmark for group activity detection that emphasizes multi-group scenarios. Note that our method is not modified to perform temporal modeling instead we apply it as-is, in a frame-wise manner. Despite the lack of temporal modeling, our method outperformed prior methods in terms of both Group mAP and Outlier mIoU. These results demonstrate the effectiveness of our method in the related task and further suggest that part-aware representations and bottom-up group reasoning not only benefit fine-grained social interaction detection, but also contribute to group activity understanding tasks.

Table 3 presents a comparison between our method and a recent MLLM, LLaVA-1.6-vicuna-7B [33], on the NVI test set. To support LLaVA, we provided ground-truth group bounding boxes and cropped the image accordingly before querying the model to identify fine-grained social interactions. Even under this favorable setup, LLaVA achieved only 37.14 AR, which is far lower than our method. Moreover, we fine-tuned LLaVA using Low-rank adaptation (LoRA) [18] on the NVI training set. We applied LoRA with rank 8 to the attention projection layers of LLaVA, enabling adaptation with a relatively small number of trainable parameters. We trained the LoRA adapter using Adam optimizer [26] with a learning rate of $1e-4$ for 15k steps. However, the result shows that LoRA fine-tuning does not yield additional gains. This suggests that even in a closed-world setting, naively fine-tuning LLaVA may not be sufficient, and additional task-specific prompt design, longer training steps, or full fine-tuning may be required.

## 4.3 In-depth analysis

We verify the effectiveness of our model through in-depth analysis on the NVI validation set.

**Impact of the pose supervision.** To validate the efficacy of our pose-guided supervision, we compare it with a variant using CLIP [40] to learn specific body-parts via text embeddings. To this end, text prompts are constructed in the form of "A photo of a person [body part]". As summarized in Table 4, our method consistently outperformed this VLM-guided variant, particularly in mR@25 and average recall (AR). It suggests that CLIP guidance is less effective in providing fine-grained spatial cues than pose estimators. We attribute this performance gap to the relatively weak spatial reasoning capabilities of current VLMs, which are primarily trained via the image-text contrastive learning.

**Impact of the proposed modules.** To address the contribution of each proposed component, we evaluate three ablated variants of our model: (1) removing the individual embedding enhancer, resulting in individual embeddings without part-aware enrichment; (2) replacing the similarity-based association with the conventional guided embedding [31, 51]; (3) removing both modules together. As shown in Table 5, removing either component leads to a noticeable drop in performance, and the degradation becomes more severe when both are excluded. It demonstrates the importance of both pose-guided part-aware representation and similarity-based bottom-up group reasoning approach.

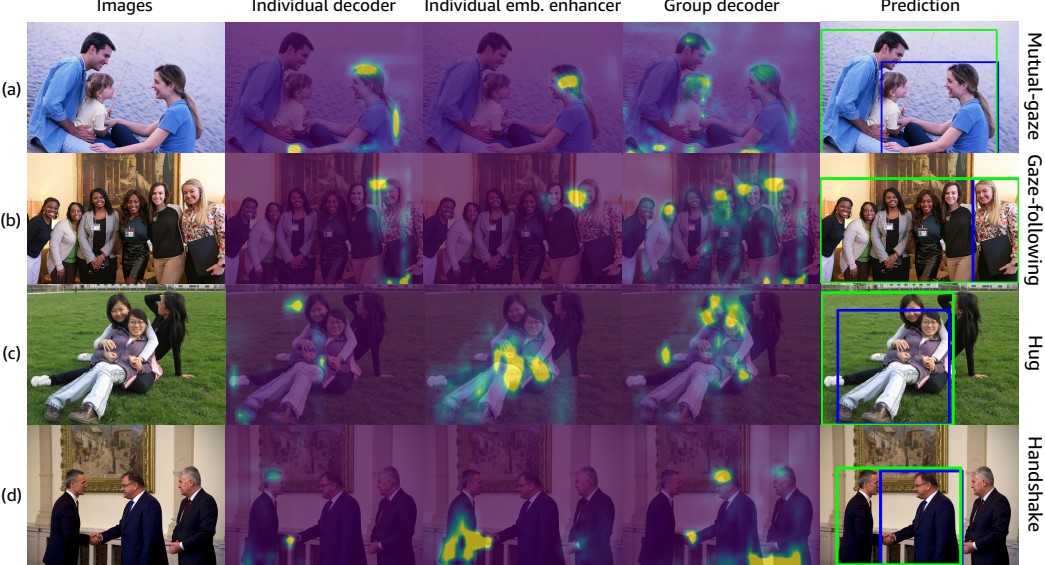

| Images | Individual decoder | Individual emb. enhancer | Group decoder | Prediction | |
|---|---|---|---|---|---|
| (a) | | | | | Mutual-gaze |
| (b) | | | | | Gaze-following |
| (c) | | | | | Hug |
| (d) | | | | | Handshake |

Figure 3: Visualizations of the cross-attention map from the individual decoder, individual embedding enhancer, and group decoder. Blue and green bounding boxes indicate individuals and groups, respectively. Predicted interaction labels are shown on the right.

**Impact of the loss function.** To assess the impact of each loss, we ablate $\mathcal{L}_{\text{part}}$, $\mathcal{L}_{\text{assn}}$, and $\mathcal{L}_{\text{loc}}$ for group localization, as described in Sec. 3.3. As shown in Table 6, removing $\mathcal{L}_{\text{assn}}$ leads to a drastic drop in performance, as the model can no longer associate individuals with their corresponding groups, underscoring the importance of similarity-based reasoning in our framework. Excluding $\mathcal{L}_{\text{loc}}$ also causes a clear performance degradation, suggesting that accurate spatial localization of group boxes is helpful. Lastly, removing $\mathcal{L}_{\text{part}}$ results in a slight performance drop, demonstrating the effectiveness of pose-guided supervision for detecting fine-grained social cues.

**Effect of the number of parts.** Table 7 summarizes the effect of the number of parts $P$. We observe that using 13 parts, which excludes redundant facial keypoints, achieves the best overall performance. Using fewer parts leads to a drop in performance. Interestingly, using all 17 keypoints does not improve performance and even results in degradation. This highlights that not all keypoints are equally useful; using overlapping or redundant parts may confuse the model and hurt model effectiveness.

**Effect of the number of queries.** Table 8 shows the effects of the number of individual queries $N_I$ and group queries $N_G$. To ensure sufficient capacity for representing multiple individuals and interactions, we set both $N_I$ and $N_G$ to be at least 24, based on the maximum number of interactions observed in the NVI training set. The best performance is achieved with 24 individual queries and 32 group queries. Unlike previous methods utilizing guided embedding that require the number of individual and group queries to be identical, our bottom-up reasoning framework with similarity-based association allows for a flexible number of queries for each component.

## 4.4 Qualitative results

**Attention visualization.** We visualize the cross-attention map of the last layer in the individual decoder, individual embedding enhancer, and group decoder for each predicted triplet in Fig 3. The individual decoder shows attention focused on spatial boundaries of the dedicated individual boxes, while the individual embedding enhancer attends to each body part of the individual; among these, we select one representative body part attention. The group decoder attends to both the spatial boundaries and regions that are essential for inferring interactions. As shown in Fig. 3 (a) and (b), the group decoder attends to the facial regions of the individuals belonging to the social group, while in Fig. 3 (d), it focuses on the hands to detect *handshake* interaction. In Fig. 3 (c), despite severe spatial overlap between individual boxes, our model successfully predicts *hug* interaction by leveraging part-aware features attending to the arms through the individual embedding enhancer.

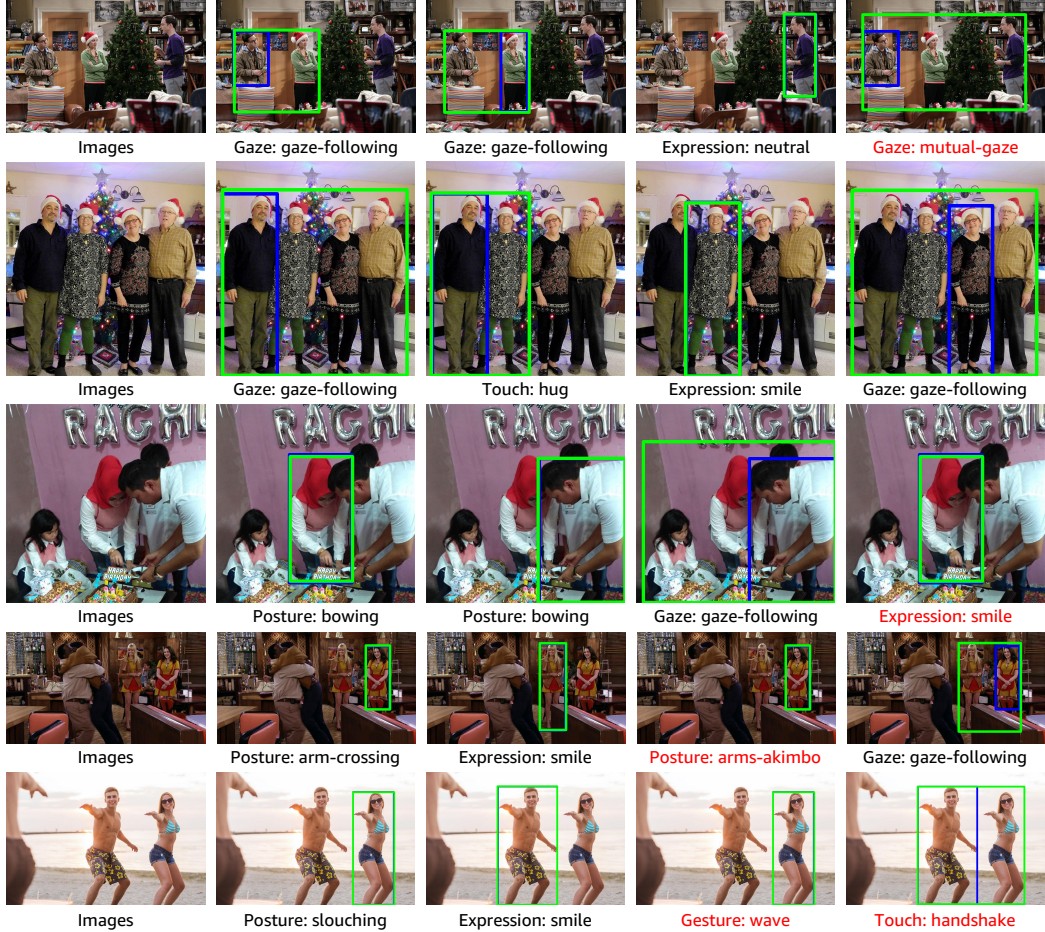

Figure 4: Qualitative results of our model on the NVI test-set. The first column shows input images, and the remaining columns visualize the predicted NVI-DET triplets. Blue and green boxes denote individuals and groups, respectively. Predicted interaction labels are presented below, where wrong predictions are highlighted in red.

**Triplet visualization.** Fig. 4 shows qualitative results of our model on the NVI test set. For each image, we select and visualize among top-10 predictions with the highest confidence scores. The results demonstrate that our model effectively localizes individuals, identifies their corresponding social groups, and recognizes fine-grained interactions through part-aware bottom-up reasoning. Even in the failure cases—for example, *arms-akimbo* in the fourth row and *wave* in the last row—the model attends to relevant part cues, such as arms and hands, resulting in plausible predictions.

## 5 Conclusion

We have presented a part-aware bottom-up group reasoning model for fine-grained social interaction detection. Our approach addresses key limitations of prior methods that rely on holistic representations of individuals and directly detect groups. By leveraging pose-guided supervision to enhance part-aware features and applying bottom-up group reasoning, our method effectively captures localized social signals that are essential for recognizing subtle and nuanced social behaviors. Extensive experiments on NVI and Café validate the effectiveness of our method. We believe this framework offers a promising direction for advancing social interaction understanding.

**Limitation.** Our model relies on the pretrained pose estimator to extract part-aware representations, which depends on external model and predefined keypoints. A valuable direction for the future would be to explore self-supervised or end-to-end approaches that can learn part-aware representations without relying on pose estimator or predefined keypoints.

## Acknowledgment

This work was supported by the NRF grant and the IITP grant funded by Ministry of Science and ICT, Korea (NRF-2021R1A2C3012728, RS-2022-II220290, RS-2022-II220926, RS-2022-II220959, RS-2019-II191906).

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

# Appendix

This appendix provides contents omitted in the main paper due to the space limit. In Sec. A, we describe the detailed architecture of the proposed model, with a particular focus on the group decoder. Sec. B presents further implementation and training details. Additional experimental results and in-depth analysis are presented in Sec. C. Qualitative comparisons with the previous state-of-the-art model, NVI-DEHR, are presented in Sec. D. More qualitative results are illustrated in Sec. E.

## A  Details of the group decoder

We provide additional details of the group decoder to clarify the design and functionality, complementing the description in Sec. 3.2. The group decoder plays a central role in our framework, as it performs part-aware bottom-up group reasoning to infer both social group configurations and their corresponding interactions. Fig. S1 illustrates detailed operations of the group decoder layers. As described in Sec. 3.2, the proposed group decoder attends to two information sources: (1) the part-aware individual embeddings $\mathbf{E}_A$ obtained through the individual decoder and individual enhancer, and (2) the encoded feature map $\mathbf{F}$. Each group decoder layer begins with multi-head self-attention among a set of learnable queries $\mathbf{Q}_G$. This is followed by a multi-head cross-attention layer where the group queries attend to the part-aware individual embeddings. Through this attention layer, the group decoder learns to associate socially relevant individuals by capturing fine-grained body part-aware cues in a bottom-up manner. Finally, as in prior work, the group queries attend to the encoded feature map to capture appearance and localize the corresponding group regions. The resulting group embeddings are then used for three purposes: (1) interaction classification, (2) group bounding box regression, and (3) computing similarity scores for determining group membership. These predictions are made through separate feed-forward heads applied to each group query output.

## B  Experimental details

We provide additional implementation details to complement those described in Sec. 4.1. Table S1 presents the specific hyperparameters used in our experiments. All experiments are conducted on four NVIDIA GeForce RTX 3090 GPUs. We implement our model using PyTorch [39] and utilize the official code repository of NVI [51], licensed under the MIT License.

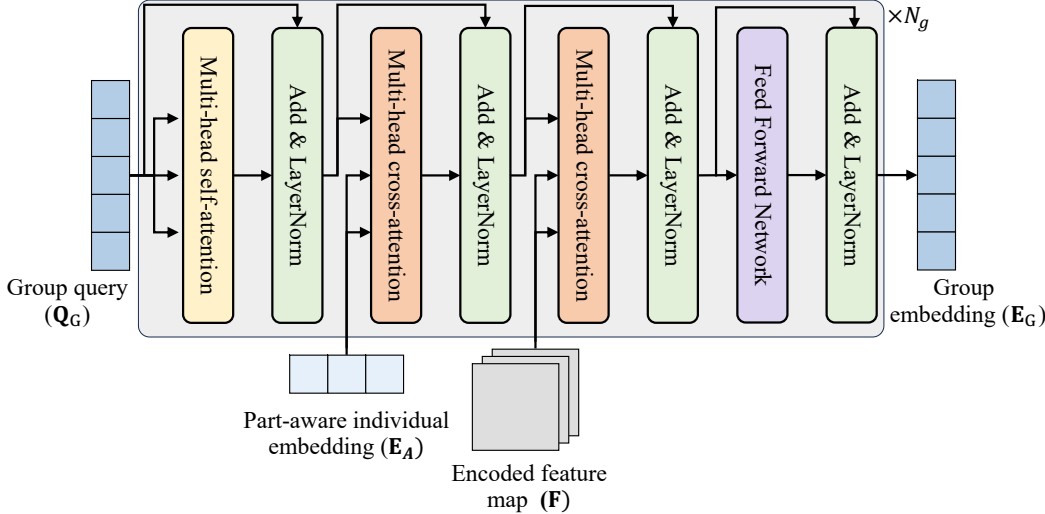

Figure S1: Detailed architecture of the group decoder.

Table S1: Hyperparameters. We provide hyperparameters used during training.

| Hyperparameters | Value |
|---|---|
| dimension in transformer $D$ | 256 |
| # of individual queries $N_I$ | 24 |
| # of group queries $N_G$ | 32 |
| # of parts $P$ | 13 |
| # of encoder layers | 6 |
| # of individual decoder layers | 3 |
| # of individual embedding enhancer layers | 3 |
| # of group decoder layers | 3 |
| Box window size ratio $\alpha$ | 0.2 |
| $\lambda_i$ | 1.0 |
| $\lambda_c$ | 2.0 |
| $\lambda_l$ | 1.0 |
| $\lambda_{\ell_1}$ | 2.5 |
| $\lambda_{GIoU}$ | 1.0 |
| $\lambda_p$ | 10.0 |
| $\lambda_a$ | 5.0 |
| # of epochs | 90 |
| batch size | 16 |
| optimizer | AdamW |
| learning rate | 1e−4 |
| backbone learning rate | 1e−5 |
| learning rate drop at epoch | 60 |
| learning rate after drop | 1e−5 |
| NMS IoU threshold $\theta$ | 0.5 |

Table S2: Impact of the pose-guided mask error.

| $\epsilon$ | mR@25 | mR@50 | mR@100 | AR |
|---|---|---|---|---|
| 0.0 | 59.43 | **76.62** | 87.43 | **74.49** |
| 0.2 | **59.90** | 75.16 | 87.84 | 74.30 |
| 0.5 | 58.52 | 76.54 | 86.64 | 73.90 |
| 1.0 | 53.26 | 74.67 | **88.34** | 72.09 |
| 2.0 | 53.89 | 74.73 | 87.94 | 72.19 |

## C  Additional experiments

**Impact of the pose error.** To verify the robustness of our method to pose errors, we perturb the outputs of the off-the-shelf pose estimator. Specifically, for each keypoint, we apply random displacements $\Delta x$ and $\Delta y$ sampled from a uniform distribution in the range $[-\epsilon \cdot s, \epsilon \cdot s]$, where $\epsilon$ controls the magnitude of the perturbation and $s$ is the window size of the pose-guided mask. Table S2 summarizes the results across varying perturbation levels. We observe that even with substantial noise (e.g., $\epsilon = 2.0$), the performance drop is marginal. These results indicate that our pose-guided supervision is robust to keypoint localization errors, and does not require highly accurate keypoint estimation to remain effective.

**Loss coefficients.** We investigate the effect of loss coefficients $\lambda_p$ and $\lambda_a$, which control the weights of the part loss $\mathcal{L}_{part}$ and the association loss $\mathcal{L}_{assn}$, respectively. As shown in Table S3, removing the association loss (i.e., $\lambda_a = 0.0$) leads to a dramatic performance drop, as the model is unable to associate individuals with their corresponding groups. While a small coefficient ($\lambda_a = 1.0$) yields suboptimal performance, the model achieves robust performance when $\lambda_a$ is in the range of 2.0 to 5.0. For $\lambda_p$, our model achieves reasonably strong performance even with zero or small loss weight, indicating that the model is not highly sensitive to this parameter. We hypothesize that this robustness arises from the model's ability to capture nuanced cues from individual embeddings through the part projection layer and individual enhancer, even with limited explicit supervision. Since our model performs well across a wide range of $\lambda_p$, from 0.0 to 10.0, we select the hyperparameter with the highest mR@25 for our final model and set $\lambda_p = 10.0$ and $\lambda_a = 5.0$.

Table S3: Impact of the loss coefficients.

| $\mathcal{L}_{\text{assn}}$ | $\mathcal{L}_{\text{part}}$ | mR@25 | mR@50 | mR@100 | AR |
|---|---|---|---|---|---|
| 0.0 | 10.0 | 30.38 | 47.20 | 64.49 | 48.32 |
| 1.0 | 10.0 | 40.18 | 57.12 | 75.53 | 57.61 |
| 2.0 | 10.0 | 57.35 | 78.31 | **88.96** | **74.88** |
| 5.0 | 0.0 | 54.32 | **78.80** | 87.60 | 73.58 |
| 5.0 | 2.0 | 59.10 | 75.09 | 88.06 | 74.08 |
| 5.0 | 5.0 | 54.48 | 75.15 | 87.37 | 72.33 |
| **5.0** | **10.0** | **59.43** | 76.62 | 87.43 | 74.49 |

Table S4: Impact of other loss coefficients.

| | $\lambda_i$ | $\lambda_c$ | $\lambda_l$ | mR@25 | mR@50 | mR@100 | AR |
|---|---|---|---|---|---|---|---|
| (1) | **1.0** | **2.0** | **1.0** | **59.43** | 76.62 | 87.43 | **74.49** |
| (2) | 0.5 | 2.0 | 1.0 | 52.78 | **78.19** | 88.94 | 73.31 |
| (3) | 2.0 | 2.0 | 1.0 | 52.88 | 74.56 | 87.49 | 71.64 |
| (4) | 1.0 | 1.0 | 1.0 | 46.32 | 73.66 | 85.85 | 68.61 |
| (5) | 1.0 | 5.0 | 1.0 | 56.83 | 75.56 | 88.78 | 73.72 |
| (6) | 1.0 | 2.0 | 0.5 | 54.31 | 75.01 | 86.88 | 72.09 |
| (7) | 1.0 | 2.0 | 2.0 | 51.17 | 73.62 | **89.00** | 71.26 |

We further investigate the effect of the other loss coefficients $\lambda_i$, $\lambda_c$ and $\lambda_l$ as suggested. Table S4 reports the performance under various combinations of these weights. First, for $\lambda_i$, comparing (1), (2), and (3), we find that increasing $\lambda_i$ to 2.0 leads the model to over-emphasize person detection, resulting in a noticeable performance drop. Next, varying $\lambda_c$, a comparison of (1), (4), and (5) shows that too small a value degrades performance, with the best result at 2.0, while 5.0 causes a slight drop. Finally, for $\lambda_l$, comparing (1), (6), and (7) reveals that the performance remains relatively stable across its values, but $\lambda_l = 1.0$ leads to the best overall results. As stated in Section 4.2 and Table S1, we select configuration (1) as our final setting.

**Triplet NMS IoU threshold.** We investigate the effect of the triplet NMS threshold in post-processing, as summarized in Table S5. As explained in Sec. 3.4, triplet NMS suppresses redundant predictions by removing overlapping triplets that satisfy these conditions: (1) their individual bounding boxes have an IoU above a threshold $\theta$, (2) their group bounding boxes also have an IoU above $\theta$, and (3) their predicted interaction class is the same. In each such group of duplicates, only the triplet with the highest confidence for the interaction class is retained. Without NMS, the performance drops approximately 10p in `all` AR. If IoU threshold $\theta$ is set too low, NMS becomes overly aggressive and removes too many predictions; if too high, duplicate predictions remain. Interestingly, we observe different trends for individual-wise interactions and group-wise interactions. For individual-wise interactions, using a high IoU threshold leaves many duplicate predictions, which negatively impacts performance. In contrast, for group-wise interactions, using a low IoU threshold aggressively removes valid predictions, resulting in performance degradation. This suggests that individual-level predictions benefit more from filtering redundant triplets, while group-level predictions are more sensitive to the loss of relevant instances. On the validation set, we find that $\theta = 0.5$ yields the best overall performance, and we use it as the default triplet NMS threshold in all experiments.

**Class-wise recall.** Table S6 compares our method and NVI-DEHR [51] in a class-wise manner. Among the 22 interaction classes, the two most underrepresented categories, beckon and palmout, exhibit notably low performance for both methods due to their rarity. Nonetheless, our method achieves notably higher recall (66.67% and 22.22%) compared to NVI-DEHR (16.67% and 11.11%), suggesting improved robustness to data imbalance. We attribute this improvement to the use of part-aware representations, which allow the model to explicitly focus on the specific body parts. Unlike holistic representations that may overlook infrequent combinations of body parts, part-aware modeling enables better generalization to individual body parts and consequently leads to more reliable detection of fine-grained interactions such as *beckon* and *palmout*, even under limited training examples for these classes.

Table S5: Effect of the triplet NMS threshold.

| IoU | individual | | | | group | | | | all |
|---|---|---|---|---|---|---|---|---|---|
| | mR@25 | mR@50 | mR@100 | AR | mR@25 | mR@50 | mR@100 | AR | AR |
| 0.1 | **58.59** | 76.08 | 85.89 | 73.52 | 56.26 | 60.80 | 62.01 | 59.69 | 69.75 |
| 0.3 | 58.03 | **78.74** | **91.64** | **76.14** | 63.12 | 70.56 | 73.66 | 69.11 | 74.22 |
| **0.5** | 55.74 | 75.65 | 88.56 | 73.32 | 69.26 | 79.18 | 84.41 | 77.62 | **74.49** |
| 0.7 | 46.43 | 74.27 | 86.33 | 69.01 | **71.70** | **83.54** | 89.43 | **81.56** | 72.43 |
| 0.9 | 43.52 | 69.22 | 82.10 | 64.95 | 71.15 | 83.07 | **89.84** | 81.36 | 69.42 |
| - | 37.51 | 61.64 | 76.29 | 58.48 | 67.49 | 77.38 | 86.13 | 77.00 | 63.53 |

Table S6: Class-wise recall (%) comparison between Ours and NVI-DEHR.

| Method | neutral | anger | smile | surprise | sadness | fear | disgust | wave | point | beckon | palm-out |
|---|---|---|---|---|---|---|---|---|---|---|---|
| **Ours** | **93.17** | 72.38 | 92.29 | 82.35 | 76.06 | 66.92 | **81.24** | **72.22** | 81.59 | 66.67 | **22.22** |
| NVI-DEHR [51] | 92.75 | **72.83** | **94.57** | 82.52 | **78.35** | **74.42** | 70.49 | 64.14 | 73.18 | 16.67 | 11.11 |

| Method | arm-crossing | leg-crossing | slouching | arms-akimbo | bowing | gaze-aversion | mutual-gaze | gaze-following | hug | handshake | hit |
|---|---|---|---|---|---|---|---|---|---|---|---|
| **Ours** | **88.19** | **87.40** | **88.03** | **87.79** | **88.81** | **70.56** | **80.67** | 78.91 | 73.74 | **74.68** | **100.00** |
| NVI-DEHR [51] | 79.79 | 85.91 | 55.13 | 70.34 | 84.28 | 60.89 | 78.34 | **83.20** | **83.73** | 74.28 | **100.00** |

## D  Qualitative comparison with previous work

Fig. S2 presents a comparison between the attention maps and predictions of our model and those of the previous state-of-the-art, NVI-DEHR [51]. Specifically, we visualize the cross-attention maps from the last layer of the interaction decoder in NVI-DEHR and the group decoder in our model. We observe distinct characteristics in the attention maps of NVI-DEHR and our model. Since NVI-DEHR adopts a guided embedding method, its attention maps tend to focus solely on the individual bounding box area (shown in blue), even for group-level interactions such as 'mutual-gaze' in (a) and 'handshake' in (b). In contrast, our model distributes attention not only within the individual but also across other relevant people with high similarity scores. In (a) and (b), our method shows strong attention on regions such as the head or hands of the group members, demonstrating the benefits of part-aware group reasoning. Examples (c) and (e), which involve individual-level interactions such as 'wave', 'point', and 'surprise', demonstrate our model's ability to focus on fine-grained body parts like the face, hands, elbows, and legs. This part-aware reasoning enables our model to detect subtle interactions that NVI-DEHR misses. In the failure case shown in (d), where both models make incorrect predictions, our model attends to hand and elbow part cues associated with interactions like 'handshake' or 'hit'—resulting in a more plausible prediction. Finally, in (g), we highlight a limitation of direct group box prediction: since NVI-DEHR does not explicitly predict group membership, it tends to attend to individuals who are spatially included in the predicted group box, even if they are not actual group members, leading to incorrect interaction predictions. In contrast, our model, while also attending to the seated person, distributes attention across relevant regions, such as the hands and faces of interacting individuals, and correctly identifies the 'mutual-gaze' interaction.

## E  Additional qualitative results

We visualize more qualitative results of our model on the NVI test set as shown in Fig. S3. For each image, we select among top-10 predictions with the highest confidence scores. These results further demonstrate the model's ability to capture diverse fine-grained social interactions. Notably, the model successfully detects posture interactions such as 'bowing', 'arm-crossing', and 'arms-akimbo', which require attention to subtle body parts. For instance, in the last column of the third row ('slouching') and the fifth row ('arm-crossing'), the model makes reasonable predictions based on the individuals' posture, even though these are not annotated in the ground-truth. This illustrates the model's sensitivity to subtle body cues even in ambiguous cases. Furthermore, it also accurately predicts gaze interactions such as 'gaze-following' and 'mutual-gaze', by attending to fine details. Our model also accurately detects various types of interactions, including touch interactions like 'hug' and 'handshake', as well as expressions such as 'smile', 'neutral', 'fear', and 'disgust', further highlighting its fine-grained perception capability.

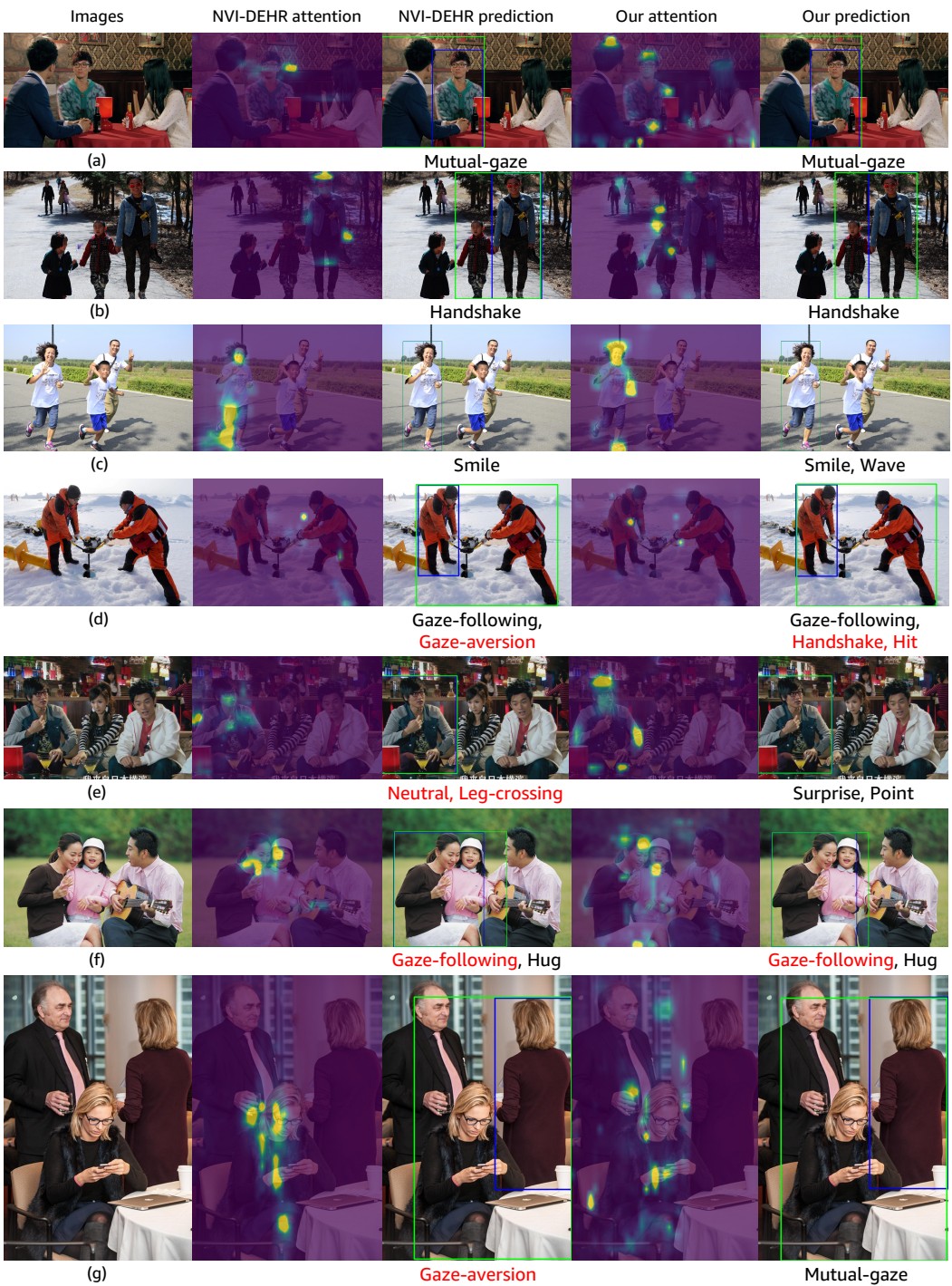

Figure S2: Qualitative comparison of cross-attention maps and predictions between NVI-DEHR [51] and our model. In the attention maps, yellow indicates high attention and purple indicates low attention value. In the prediction result, blue and green bounding boxes denote individuals and groups, respectively. Incorrect predictions are highlighted in red.

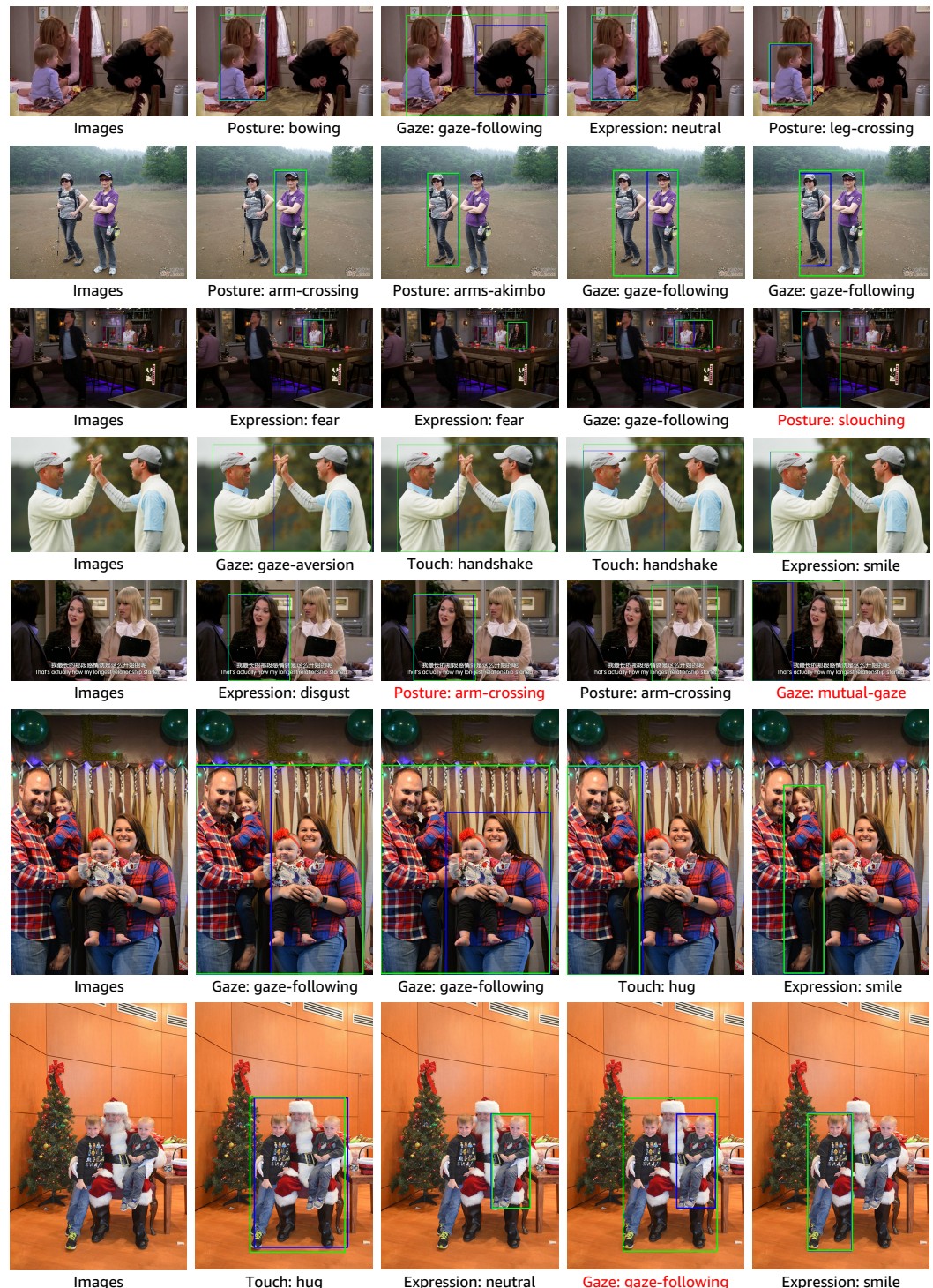

Figure S3: Additional visualizations of our model's predictions on the NVI test set. The first column shows the input image, while the remaining columns visualize the predicted triplets. Blue and green bounding boxes denote individual and groups, respectively. Predicted interaction classes are listed below, with incorrect predictions highlighted in red.

