# OpenReview forum: "Part-Aware Bottom-Up Group Reasoning for Fine-Grained Social Interaction Detection"
_NeurIPS.cc/2025/Conference — NeurIPS 2025 poster_

### Official Review · Reviewer_UmiK · 2025-06-20

**Clarity:** 3
**Significance:** 3
**Originality:** 2
**Rating:** 4
**Confidence:** 3

**Summary:**

This paper studies non-verbal interaction detection, which detects the interaction between a person and a group in the image. The proposed method is inspired by DETR, where a CNN backbone and a transformer encoder extract features, with two decoders for person detection and interaction recognition. To better recognize the fine-grained human parts for interaction recognition, this method also uses ViTPose to extract human keypoints, which are then used as L_{part} to regularize the attention maps. The proposed method achieves SOTA results on NVI-DET benchmark. Additional ablation analysis are conducted to validate the design of each proposed component.

**Questions:**

Please refer to weakness section for my questions.

**Ethical Concerns:**

["NO or VERY MINOR ethics concerns only"]

**Final Justification:**

Thanks the authors for the detailed rebuttal. Although there are some weaknesses such as limited novelty as less comprehensive experiments, I think the authors addressed my questions during the rebuttal period, I thus raise my rating of this paper.

However, I do agree with other reviewers that the evaluation on the Cafe dataset is somewhat wired. So I lowered my confidence of this paper.

**Limitations:**

The authors do discussed the limitations in the paper, which mainly focuses on the utilization of the external pose detection model. However, I think there are still other limitations of the proposed method and the studied task that are not discussed in the paper. For example

* the ambiguility of the interactions, which may stem from the inherent difference between the image and text modalities.

* the closed-world setting for interaction recognition, where interaction types unseen during training are not recognizable during inference. This can be somehow solved by combining a visual understanding model with modern LLMs, but can not be handled with the model proposed in this paper.

* the privacy issues.  I image with sufficient human information should be given to the model for recognition, which requires the human information.

I appreciate the efforts of the authors made in this work and understand that these points can not be handled with a single or several works. But discussing them in the paper could make it more coherent.

**Quality:**

3

**Strengths And Weaknesses:**

Strengths:
* The problem studied in this paper is important for some real world applications.
* The proposed method achieved SOTA performance on benchmarks.
* The ablation studies are generally comprehensive to validate design of each proposed component.
* The paper is well-written and easy-to-follow. Every proposed component are explained clearly.

Weaknesses:
* The task studied in this paper, NVI-DET, is very similar to HOI-detection, except that latter detects <human, object, interaction> while this paper studies <human, group, interaction>. Although the authors discussed in this paper that the NVI-DET requires subtle human clues for interaction recognition, the same also applies to HOI-DET. To me, it is unclear how this task significantly differs from the existing HOI-DET task, which has been studied for several years in the previous literature.

* The proposed achitecture is also very similar to a previous work on HOI-DET, i.e., GEN-VLKT [1]. Both models applies the idea of DETR, and adopt two decoders, where one decoder for instance detection while the other for interaction detection&recognition. Although the utilization of human pose seems to be a new contribution to the NVI-DET task, the same idea has been extensively utilized in HOI-DET tasks [2,3]. As such, I think the novelty of this paper is quite limited. The authors could further elaborate and discuss on the difference between this work and existing HOI literacture, like why the human information in HVI-DET is different from that in HOI-DET.

* The proposed method can only handle closed-world settings, where interactions unseen during training and not recognizable during inference. However, this is already achievable in a previous work [1]. In the era of various pretrained and large models (e.g., CLIP, various MLLMs), it constitutes a deficiency of the model to be able to handle seen interactions only.


References:

[1] Liao et. al. "GEN-VLKT: Simplify Association and Enhance Interaction Understanding for HOI Detection". In CVPR 2022

[2] Yu et. al. "Exploring Pose-Aware Human-Object Interaction via Hybrid Learning", In CVPR 2024

[3] Wan et. al. "Pose-aware Multi-level Feature Network for Human Object Interaction Detection". In CVPR 2020


 Overall, I am looking forward to the rebuttals from the authors and would like to change my ratings after seeing the rebuttal.

---

> ### Author Rebuttal · Authors · 2025-07-31
>
> ## How NVI-DET differs from HOI-DET task?
> > The task studied in this paper, NVI-DET, is very similar to HOI-detection, except that latter detects <human, object, interaction> while this paper studies <human, group, interaction>. Although the authors discussed in this paper that the NVI-DET requires subtle human clues for interaction recognition, the same also applies to HOI-DET. To me, it is unclear how this task significantly differs from the existing HOI-DET task, which has been studied for several years in the previous literature.
>
> We agree that NVI-DET shares structural similarities with human-object interaction detection (HOI-DET).
> However, Wei et al. [44] state that
> “Unlike HOI-DET that mainly focuses on recognizing actions between human-object pairs, NVI-DET aspires to comprehensively interpret the full spectrum of communicative nonverbal signals, whose patterns are generally more subtle, ambiguous, and involving multiple persons”.
> In this regard, NVI-DET requires hierarchical reasoning over multiple individuals and their group-level interactions, making it fundamentally more complex than HOI-DET.
>
> ## Limited novelty
> > The proposed achitecture is also very similar to a previous work on HOI-DET, i.e., GEN-VLKT [28]. Both models applies the idea of DETR, and adopt two decoders, where one decoder for instance detection while the other for interaction detection&recognition. Although the utilization of human pose seems to be a new contribution to the NVI-DET task, the same idea has been extensively utilized in HOI-DET tasks [R2, 46]. As such, I think the novelty of this paper is quite limited. The authors could further elaborate and discuss on the difference between this work and existing HOI literacture, like why the human information in HVI-DET is different from that in HOI-DET.
>
> As pointed out by the reviewer, DETR-based architectures with separate decoders for instance and interaction prediction have been widely used in human interaction understanding [22, 28, 44].
> For instance, GEN-VLKT [28] and NVI-DEHR [44] form interaction queries by fusing human-object embeddings.
> In contrast, our method first detects individuals and then introduces separate group queries that attend to individual embeddings and dynamically aggregate relevant individuals into a group representation.
> This design enables joint reasoning of group composition and interaction recognition, allowing social groups to emerge based on inter-personal relations rather than directly predicted from image features.
> Furthermore, we enhance individual representations through a part-aware representation learning module supervised by pose guidance, which enables more fine-grained reasoning over nonverbal cues.
> Unlike HOI-DET methods using human pose as an explicit input during both training and inference [R2, 46], our work leverages human pose only for training as a privileged information to learn part-aware representations.
> This design improves both generalizability and efficiency, as the model does not depend on pose estimators at inference.
> We appreciate the reviewer's suggestion and will cite Wan et al. [R2] in addition to the already-cited Wu et al. [46].
>
> ## Limitation of the closed-world setting and the use of LLMs
> > The proposed method can only handle closed-world settings, where interactions are unseen during training and not recognizable during inference. However, this is already achievable in a previous work [28]. In the era of various pretrained and large models (e.g., CLIP, various MLLMs), it constitutes a deficiency of the model to be able to handle seen interactions only. This can be somehow solved by combining a visual understanding model with modern LLMs, but can not be handled with the model proposed in this paper.
>
> We appreciate the reviewer’s insight on extending our work towards open-world settings and the potential of leveraging MLLMs.
> As noted by the reviewer, VLMs and MLLMs have shown impressive performance across various tasks.
> However, for complex tasks such as NVI-DET and HOI-DET—which require pair-wise detection and interaction recognition—the performance of MLLMs without fine-tuning remains limited.
> To empirically examine this, we utilized a recent MLLM, LLaVA-1.6-vicuna-7B [R4], on the NVI test set.
> We first prompted the model to directly output <individual, group, interaction> triplets, but this did not yield meaningful results.
> To support LLaVA, we provided ground-truth group bounding boxes and cropped the image accordingly before querying the model to identify fine-grained social interactions.
> Even under this favorable setup, LLaVA achieved only 31.88 AR, which is far lower than our method (Table R2).
> This suggests that even in a closed-world setting, MLLMs struggle with fine-grained reasoning without task-specific tuning.
> We acknowledge that our work is limited to the closed-world setting, and we agree that extending it to the open-world setting is an important future direction.
> We chose to focus on the closed-world setting because we believe that fine-grained social interaction detection still remains challenging and underexplored even in this setup.
> In the future work, we plan to explore incorporating VLMs to enable more generalizable social interaction understanding.
>
> **Table R2: Comparison with LLMs.**
> | Method | mR@25 | mR@50 | mR@100 | AR    |
> |--------|-------|-------|--------|-------|
> | **Ours**   | **63.59** | **80.62** | **91.34**  | **78.52** |
> | LLaVA  |  13.44 | 28.98 | 53.21  |  31.88 |
>
> ## Ambiguity of the interactions
> > The ambiguity of the interactions, which may stem from the inherent difference between the image and text modalities.
>
> We thank the reviewer for highlighting this important point.
> We agree that ambiguity is inherent in social interactions.
> This is also why NVI-DET adopted recall as the primary evaluation metric, since precision is less appropriate given the possibility of multiple plausible interactions that may not be exhaustively annotated in the dataset.
>
> ## Privacy issue
> > Image with sufficient human information should be given to the model for recognition, which requires the human information.
>
> We appreciate the reviewer’s concern regarding privacy.
> Recognizing fine-grained social interactions inevitably requires detailed human-centric visual cues, many of which can involve sensitive personal information.
> The creators of the NVI dataset [44] have already addressed this concern explicitly in their paper.
> As they stated:
> *“There is a risk that someone could use it for malicious purposes, e.g., widespread surveillance, invasion of privacy, and potential abuse of personal information. Therefore, we strongly advocate for the well-intended application of the proposed method, while simultaneously underscoring the importance of employing the dataset in a responsible and ethical manner”.*
> We will include a discussion on the ethical considerations in the revised manuscript.
>
> ---
> [R2] Wan et al., Pose-aware Multi-level Feature Network for Human Object Interaction Detection, ICCV 2019.
>
> [R4] Liu et al., Visual Instruction Tuning, NeurIPS 2023.

---

> ### Comment · Reviewer_UmiK · 2025-08-01
>
> Thanks for the response.
>
> * Under the question "How NVI-DET differs from HOI-DET task?", the authors claim that HVI focuses more on group-level activity, but this has already been studied in the literature of HOI-DET like [1].
>
> * I think one merit of the proposed method is that it only requires human part during training, making it less cumbersome for inference. I suggest the authors to further hightlight this in the revision.
>
> * I like the experiments which test LLaVA on the NVI test set. A more interesting experiments would be to perform SFT on LLaVA on the NVI training set and test it on the NVI test split. This may be out of the scope of this work since it may require proper feature engineering and remains unexplored. But I think it could be a promising future direction.
>
> [1] Liu et al. Interactiveness Field of Human-Object Interactions. In CVPR 2022.

---

> > ### Author Response · Authors · 2025-08-02
> >
> > We appreciate your careful and timely review of our responses.
> > - Thank you for pointing out interesting work [R9]. This work models **interactiveness paring** between a human and an object in a binary pairwise manner. While both this work and ours aim to understand interactions beyond simple detection, our focus is on **group-level** interactions beyond binary pairwise interactions—for example, four-person gaze-following, as illustrated in the second row of Fig. 4 in the main paper. We will clarify this distinction and will discuss this work and its relation to our work in the revised version.
> > - Thank you for your constructive feedback and for recognizing the merit of our method. We will make up this advantage in the revised manuscript.
> > - We also appreciate your insightful suggestion regarding supervised fine-tuning of MLLMs. We believe it is meaningful to examine whether the extensive prior knowledge of MLLMs can benefit tasks like NVI-DET. We have initiated preliminary experiments in this direction and hope to share some findings within the discussion period.
> >
> > ---
> > [R9] Liu et al. Interactiveness Field of Human-Object Interactions, CVPR 2022.

---

> > ### Author Response · Authors · 2025-08-09
> > **Supervised fine-tuning of LLaVA**
> >
> > Dear reviewer UmiK,
> >
> > Thank you again for your valuable suggestion regarding supervised fine-tuning (SFT) of LLaVA on the NVI training set.
> > We sincerely apologize for the delay in sharing the experimental results, but we hope the findings are still of interest and contribute to the ongoing discussion.
> >
> > As mentioned in our earlier response, we found your comment insightful and initiated preliminary experiments by fine-tuning LLaVA using Low-rank adaptation (LoRA) [R10] on the NVI training set.
> > To keep the setup lightweight and computationally efficient, we applied LoRA with rank 8 to the attention projection layers of LLaVA-1.6-vicuna-7B, enabling adaptation with a relatively small number of trainable parameters.
> > We trained the LoRA adapter using ADAM optimizer with a learning rate of $1\times 10^{-4}$ for 15k steps.
> >
> > As shown in Table R7, our initial results show that while prompt tuning leads to a meaningful improvement over the baseline (AR 31.88 to 37.14), LoRA fine-tuning does not yield additional gains (AR 33.81).
> > This suggests that naively fine-tuning LLaVA may not be sufficient, and additional task-specific prompt design, longer training steps, or full fine-tuning—may be required.
> > Exploring such direction may yield even stronger models for this task, and we agree this represents a promising avenue for future work.
> >
> > Please let us know if you have any further questions or suggestions, we would be happy to follow up.
> >
> >
> > **Table R7: Comparison with LLaVA and LLaVA-LoRA.**
> > | Method | mR@25 | mR@50 | mR@100 | AR    |
> > |--------|-------|-------|--------|-------|
> > | **Ours**   | **63.59** | **80.62** | **91.34**  | **78.52** |
> > | LLaVA  |  13.44 | 28.98 | 53.21  |  31.88 |
> > | LLaVA (New prompt)  |  21.09 | 36.75 | 53.59  | 37.14 |
> > | LLaVA-LoRA (15K steps)  |  17.40 | 32.12 | 51.93  |  33.81 |
> >
> > ---
> > [R10] Hu et al., LoRA: Low-Rank Adaptation of Large Language Models, ICLR 2022.

---

### Official Review · Reviewer_iWA3 · 2025-06-24

**Clarity:** 3
**Significance:** 3
**Originality:** 3
**Rating:** 4
**Confidence:** 4

**Summary:**

This paper proposes a part-aware bottom-up framework for fine-grained social interaction detection by modeling subtle cues (e.g., facial expressions, gaze) from body parts. Unlike existing holistic approaches, the proposed method enhances individual features with part-level details, then infers groups through similarity reasoning incorporating both spatial relations and social signals. Experiments on NVI dataset demonstrate state-of-the-art performance in detecting interaction-based groups

**Questions:**

(1)	The potential issue is that the model does not clarify the expected range of interacting group members. If too many people are involved, pose estimation may suffer from occlusions, which could reduce the accuracy of the pose-guided mask.
(2)	In some tasks related to group activity recognition (your Related Work), there are several classic datasets, such as the Volleyball [1] and Collective Activity Dataset [2]. These datasets focus more on group actions—can they be considered as representations of social interactions?
[1] Mostafa S. Ibrahim, Srikanth Muralidharan, Zhiwei Deng, Arash Vahdat, and Greg Mori. 2016. A hierarchical deep temporal model for group activity recognition. In Proceedings of the IEEE Conference on Computer Vision and Pattern Recognition. 1971–1980.
[2] Wongun Choi, Khuram Shahid, and Silvio Savarese. 2009. What are they doing?: Collective activity classification using spatio-temporal relationship among people. In Proceedings of the 2009 IEEE 12th International Conference on Computer Vision Workshops, ICCV Workshops. 1282–1289.

**Ethical Concerns:**

["NO or VERY MINOR ethics concerns only"]

**Final Justification:**

Thanks for authors’ response and I will maintain my orignal rating score.

**Limitations:**

(1)	In the introduction, when social interaction is first mentioned, it is advisable to include a definition, as referenced in Marco Cristani et al. “Social interaction discovery by statistical analysis of F-formations. (BMVC 2011)”.
(2) It would be beneficial if the authors could provide a comparison of the training and/or inference time between their proposed method and existing approaches. Highlighting the computational efficiency or runtime advantage would further demonstrate the practicality and scalability of the method in real-world applications.
(3) In the supplementary material, the authors provide an analysis of the hyperparameters λₚ and λₐ. However, it would be helpful to also include a discussion of the other loss weights, such as λᵢ, λ_c, and λ_l. Including these would offer a more complete understanding of the model's sensitivity to different components of the overall loss function.

**Quality:**

3

**Strengths And Weaknesses:**

Strengths And Weaknesses:
•	Strengths: This paper addresses the problem of social interaction detection by focusing on fine-grained limb features. Its core strengths lie in a bottom-up interaction modeling approach, which progresses from individuals to pairwise relationships and then to groups, ensuring that group structures naturally emerge from interpersonal interactions (in contrast to traditional methods that directly detect group structures). Additionally, it employs part-aware feature enhancement by extracting local features from facial, arm, and leg regions based on pose estimation, accurately capturing fine-grained interaction semantics that are easily confused.
•	Weaknesses:
(1)	One limitation lies in the clarity of Figure 1, particularly in the "Bottom-up group reasoning" section. The visual representation of the embedding process, especially the part involving pose-based interaction, is not clearly aligned with the textual description in the manuscript.  Improving the figure’s clarity and its correspondence with the written explanation would enhance the overall comprehensibility of the proposed approach.
(2)	The transition introduced in line 107 ("Unlike this approach...") appears abrupt and lacks a clear logical bridge from the preceding discussion. While the paragraph describes the strengths of prior work such as [54], it does not provide a critical analysis of its limitations. A smoother transition that explicitly contrasts the proposed method with the weaknesses of existing approaches would strengthen the narrative flow and clarify the novelty of the current work.
(3) In Figure 3(a), the illustrated example appears to involve both gaze and hug interactions simultaneously. It would be helpful if the authors could clarify how such cases are handled in the dataset annotation or during model inference. Specifically, how are multiple co-occurring interaction types defined, labeled, or distinguished—either in the data or by the algorithm?

---

> ### Author Rebuttal · Authors · 2025-07-31
>
> ## Clarify of Fig. 1
> > The visual representation of the embedding process, especially the part involving pose-based interaction, is not clearly aligned with the textual description in the manuscript.
>
> Thank you for the constructive comment. As noted by the reviewer, the current figure incorrectly depicts the concatenation of the part-specific query $Q_P$ and the part embedding $E_P$, which may lead to confusion.
> In accordance with the textual explanation in the manuscript, the part embedding $E_P$ should be concatenated with the corresponding individual embedding $E_I$—not with the part-specific query $Q_P$.
> We will correct this misrepresentation in the revision to ensure consistency between the figure and text, thereby improving the overall clarity of the proposed bottom-up group reasoning framework.
>
> ## Limitation of prior work
> > While the paragraph describes the strengths of prior work such as [44], it does not provide a critical analysis of its limitations.
>
> We will revise the paragraph to more clearly contrast the proposed method with the limitations of prior work.
> Specifically, we will clarify that prior work [44] detects social groups directly without explicitly modeling inter-person relations—a limitation that becomes particularly problematic when detecting interactions like gaze, which often occur between spatially distant individuals.
> We will also point out that prior work relies on holistic person representations, thereby overlooking body part-level cues that are essential for distinguishing visually similar but semantically distinct interactions (e.g., *wave* vs. *point*, or *mutual-gaze* vs. *gaze-following*), as noted in the introduction (Line 40-52).
>
> ## Handling co-occurring multiple interactions
> > It would be helpful if the authors could clarify how such cases are handled in the dataset annotation or during model inference. Specifically, how are multiple co-occurring interaction types defined, labeled, or distinguished—either in the data or by the algorithm?
>
> Both the dataset annotation and our model are designed to handle the co-occurrence of multiple interactions.
> Following NVI-DEHR [44], we formulate interaction recognition as a multi-label classification task: for each detected group, the model outputs a probability vector over 22 interaction classes, indicating the presence of each interaction.
> For instance, in the fourth row of Figure S3 in the supplementary material, one group simultaneously exhibits both *gaze-aversion* and *handshake* interactions.
> Moreover, the dataset allows one person to belong to multiple groups simultaneously, allowing overlapping social groups.
>
> ## Impact of accuracy of the pose-guided mask
> > The potential issue is that the model does not clarify the expected range of interacting group members. If too many people are involved, pose estimation may suffer from occlusions, which could reduce the accuracy of the pose-guided mask.
>
> We agree that the accuracy of the pose-guided mask could be degraded when many people are present in the scene, due to occlusions.
> We found that, in the training set, the average number of individuals involved in a group interaction is $2.02$, with some groups involving as many as $16$ people.
> However, our method is robust to such errors because the pose-guided mask provides soft supervision that guides the model to attend to the expected regions of each body part rather than enforcing exact localization.
> To verify the robustness of our method to pose errors, we perturb the outputs of the off-the-shelf pose estimator (as ground-truth pose annotations are not available).
> Specifically, for each keypoint, we apply random displacements $\Delta x$ and $\Delta y$ sampled from a uniform distribution in the range $[-\epsilon \cdot s, \epsilon \cdot s]$, where $\epsilon$ controls the magnitude of the perturbation and $s$ is the window size of the pose-guided mask (Line 214).
> Table R5 summarizes the results across varying perturbation levels.
> We observe that even with substantial noise, e.g., $\epsilon=2.0$, the performance drop is minor.
> These results indicate that our pose-guided supervision is robust to moderate keypoint localization errors, and does not require highly accurate keypoint estimation to remain effective.
>
> **Table R5: Impact of the pose-guided mask error.**
> | ε     | mR@25 | mR@50 | mR@100 | AR    |
> |--------|--------|--------|---------|--------|
> | 0.0    | 59.43  | 76.62  | 87.43   | **74.49**  |
> | 0.2    | 59.90  | 75.16  | 87.84   | 74.30  |
> | 0.5    | 58.52  | 76.54  | 86.64   | 73.90  |
> | 1.0    | 53.26  | 74.67  | 88.34   | 72.09  |
> | 2.0    | 53.89  | 74.73  | 87.94   | 72.19  |
>
> ## Applicability to group activity understanding
> > In some tasks related to group activity recognition, there are several classic datasets, such as the Volleyball [R6] and Collective Activity Dataset [R7]. These datasets focus more on group actions—can they be considered as representations of social interactions?
>
> We thank the reviewer for this insightful question.
> We agree that group activity can be considered as a form of social interaction.
> However, most group activity benchmarks—such as Volleyball [R6], CAD [R7], or Café [22]—focus on coarse-grained interactions (e.g., *walking*, *queuing*, *fighting*) rather than the fine-grained social interaction (e.g., *gesture*, *gaze*, *facial expression*) that our work primarily targets.
> Nevertheless, to further validate the applicability of our method beyond fine-grained interaction detection, we conducted additional experiments on Café [22], a recent benchmark that involves multiple group activities in complex scenes (Table R3).
> The results demonstrate that the proposed part-aware representation and bottom-up group reasoning are beneficial even in understanding group activities, demonstrating the broader applicability of our method.
>
>
> **Table R3: Comparison with the previous methods on Café detection-based setting.**
> | Method        | Group mAP$_{1.0}$ (view) | Group mAP$_{0.5}$ (view) | Outlier mIoU (view) | Group mAP$_{1.0}$ (place) | Group mAP$_{0.5}$ (place) | Outlier mIoU (place) |
> |---------------|----------------------|------------------------|----------------------|------------------------|------------------------|------------------------|
> | Joint [9]    | 9.14                 | 31.83                  | 42.93                | 6.08                   | 18.43                  | 2.83                   |
> | JRDB-base [10] | 12.63                | 35.53                  | 31.85                | 8.15                   | 22.68                  | 33.03                  |
> | HGC [39]       | 6.77                 | 31.08                  | 57.65                | 4.27                   | 24.97                  | 57.70                  |
> | Cafe-base [22] | 14.36              | 37.52                | 63.70              | 8.29                 | 28.72                | 59.60                |
> | **Ours**      | **18.23**            | **46.88**              | **67.62**            | **10.65**              | **39.03**              | **63.60**              |
> ## Definition of the social interaction
> We thank the reviewer for the helpful suggestion.
> We will include a formal definition of social interaction and cite the paper [R8] when it is first introduced in Section 1 as recommended.
> ## Comparison of the training and/or inference time
> We thank the reviewer for useful comments.
> We compare the inference time of our model and NVI-DEHR using $224 \times 224$ resolution inputs.
> Our method achieves an inference speed of **26.62ms**, which is slightly faster than NVI-DEHR’s **27.07ms**.
>
> ## Discussion of the other loss weights
> > It would be helpful to also include a discussion of the other loss weights, such as $\lambda_\text{i}$, $\lambda_\text{c}$, and $\lambda_\text{l}$.
>
> We investigated the effect of the other loss coefficients $\lambda_\text{i}$, $\lambda_\text{c}$ and $\lambda_\text{l}$ as suggested.
> Table R6 reports the performance under various combinations of these weights.
> First, for $\lambda_\text{i}$, comparing (1), (2), and (3), we find that increasing $\lambda_\text{i}$ to $2.0$ leads the model to overly emphasize person detection, resulting in a noticeable performance drop.
> Next, varying $\lambda_\text{c}$, a comparison of (1), (4), and (5) shows that too small a value degrades performance, with the best result at $2.0$, while $5.0$ causes a slight drop.
> Finally, for $\lambda_\text{l}$, comparing (1), (6), and (7) reveals that the performance remains relatively stable across its values, but $\lambda_\text{l}=1.0$ leads to the best overall results.
> As stated in Section 4.2 and Table S1, we select configuration (1) as our final setting.
>
> **Table R6: Impact of the other loss coefficients.**
> |      | $λ_i$  | $λ_c$  | $λ_l$  | mR@25     | mR@50     | mR@100    | AR      |
> |------|-----|-----|-----|-----------|-----------|-----------|---------|
> | (1)  | 1.0 | 2.0 | 1.0 | 59.43 | 76.62 | 87.43     | **74.49** |
> | (2)  | 0.5 | 2.0 | 1.0 | 52.78     | 78.19     | 88.94    | 73.31   |
> | (3)  | 2.0 | 2.0 | 1.0 | 52.88    | 74.56      | 87.49     | 71.64   |
> | (4)  | 1.0 | 1.0 | 1.0 | 46.32     | 73.66     | 85.85     | 68.61   |
> | (5)  | 1.0 | 5.0 | 1.0 | 56.83   | 75.56   | 88.78   | 73.72 |
> | (6)  | 1.0 | 2.0 | 0.5 | 54.31     | 75.01     | 86.88     | 72.09   |
> | (7)  | 1.0 | 2.0 | 2.0 | 51.17     | 73.62     | 89.00 | 71.26   |
>
> ---
> [R6] Ibrahim et al., A hierarchical deep temporal model for group activity recognition, CVPR 2016.
>
> [R7] Choi et al., What are they doing?: Collective activity classification using spatio-temporal relationship among people, ICCVW 2009.
>
> [R8] Cristani et al., Social interaction discovery by statistical analysis of F-formations, BMVC 2011.

---

> ### Author Response · Authors · 2025-08-06
> **A gentle reminder**
>
> Dear reviewer iWA3,
>
> Thank you again for the time and effort you have dedicated to reviewing our submission.
> We have carefully addressed your comments in our rebuttal, and we would greatly appreciate it if you could share any additional thoughts or let us know if there are any remaining concerns.
> Your feedback would be invaluable in helping us improve our work.
> Thank you again for your thoughtful review and consideration.
>
> Best regards,
>
> Authors

---

> > ### Comment · Area_Chair_vAd6 · 2025-08-06
> > **Please check the authors' rebuttal!**
> >
> > Dear reviewer iWA3,
> > Please check the authors' rebuttal, engage in a discussion if there are remaining concerns and adjust your scores and review accordingly.
> > Thanks,
> > AC

---

> ### Comment · Reviewer_iWA3 · 2025-08-08
>
> Thanks for your response. My concerns have been well addressed.

---

### Official Review · Reviewer_tcQL · 2025-07-03

**Clarity:** 3
**Significance:** 2
**Originality:** 2
**Rating:** 4
**Confidence:** 4

**Summary:**

This paper presents a bottom up reasoning framework for group behaviour reasoning. Particularly, the method starts from the body parts of the individuals in the images, and is composed of dedicated decoders for individuals and groups and an association module for group behaviour reasoning such as looking at each other or hand shaking. The proposed methodology is evaluated on one dataset.

**Questions:**

- What is the contribution to the domain of machine learning?
- What are the authors focusing on images, which is limited? Today LLMs are pretty good at reasoning about images, but they are failing on videos. How do the authors compare their method with LLM-based reasoning methods?
- Can this method be applied to other datasets and downstream tasks?

**Ethical Concerns:**

["NO or VERY MINOR ethics concerns only"]

**Final Justification:**

The reviewer thanks the authors for their hard work and their detailed response.

While the reviewer acknowledges that this paper addresses an important topic in advancing our understanding of automatic social reasoning, they believe it is not a good fit for this venue, which primarily emphasizes contributions to machine learning methodology. Although the use of pose-guided approaches is relevant, it is not a novel concept and has been extensively explored in other contexts. Its application to this particular problem, while interesting, does not represent a sufficiently innovative contribution for NeurIPS.

Additionally, the paper's evaluation is limited to a single dataset with a narrow range of social interaction classes, all within the image domain. This further constrains its generalisability and impact.

During the rebuttal, the authors provided additional results on the Café dataset. However, the rationale for the baseline modelling approach, as compared to temporal modelling, remains unclear. The reviewer is therefore uncertain about the final version of this paper.

Overall, the paper is technically sound and well-written. However, given the considerations above, it does not meet the bar for acceptance at NeurIPS in the reviewer's opinion. Therefore, the reviewer maintains their decision.

**Limitations:**

Limitations are adequately discussed, except for the weaknesses outlined above. Additionally, visualisation results are helpful in both in the paper and the supplementary material.

**Paper Formatting Concerns:**

The paper is written and organised well. Only concern is that it is not clear, until the experimental results, the output of the method. In other words, what is the goal of social reasoning? This becomes clear with the visual results. However, if the problem definition can be introduced clearly in the beginning, it would be helpful for the readers.

**Quality:**

2

**Strengths And Weaknesses:**

The reviewer appreciates the hard work of the authors. The strengths and weaknesses are summarised below.

Strengths:
- Group behaviour understanding is still a growing topic. The majority of work treats gaze communication behaviour and activity recognition separately. This paper offers a unified framework for reasoning.
- The results demonstrate the potential of the approach.

Weaknesses:
- The technical novelty is limited. The proposed architecture is a very typical architecture, feature extraction, individual and group modelling, and association module. It is not clear the main contribution to the machine learning domain. Part-aware representation seem to be only contribution, which is incremental.
- The proposed method focuses on images only, which has limited applications.
- The proposed method is tested on one dataset only, which is substandard. The reviewer understands that the datasets for this problem might be limited. However, the authors could test this on gaze communication datasets such as GPStatic. Alternatively, there is a dataset for group activity recognition called JRDB dataset, https://jrdb.erc.monash.edu.
- Following the previous comment, the experimental results are limited. For instance the authors could have implemented stronger baselines to compare their work, for instance, leaderboard methods on the JRDB dataset.
- The paper could also benefit further analysis. For instance, one of the problems with the group activity datasets is the data imbalance problem. It would be good to see the results over different classes to see if the method is performing equally well and where are the limitations.

---

> ### Author Rebuttal · Authors · 2025-07-31
>
> ## Contribution to the domain of machine learning (technical novelty)
> > The technical novelty is limited. The proposed architecture is a very typical architecture, feature extraction, individual and group modelling, and association module. It is not clear the main contribution to the machine learning domain. Part-aware representation seem to be only contribution, which is incremental.
>
> We agree that our framework is based on a commonly used architecture.
> However, we argue that the significance of our work stems from how we leverage these standard components to address the unique challenges of fine-grained social interaction detection, which remains underexplored despite its importance.
> Our key contribution lies in introducing part-aware representations that are learned using human pose labels as privileged information [R1].
> In other words, our model leverages human pose only for training, unlike HOI-DET methods [R2, 46] that explicitly use human pose as an additional input for both training and inference.
> This design allows the model to benefit from fine-grained supervision while maintaining efficient inference without requiring additional inputs.
> To the best of our knowledge, such an approach using pose-guided supervision has not been studied in the literature of fine-grained social interaction detection.
>
> ## Why are focusing on images?
>
> We focus on images in this work because existing research on interaction detection—such as NVI-DET and HOI-DET—is predominantly conducted in the image domain, with widely used benchmarks providing image-level annotations.
> We however agree that extending this line of work to the video domain remains an important future direction and could further enhance the applicability of these methods.
>
> ## Comparison with VLMs or LLMs
> > Today LLMs are pretty good at reasoning about images, but they are failing on videos. How do the authors compare their method with LLM-based reasoning methods?
>
> We thank the reviewer for pointing out this interesting direction.
> Recent studies in HOI-DET [25,28] have indeed explored the use of VLMs or LLMs for interaction understanding.
> However, these models are rarely used for direct reasoning about visual inputs, and are instead often employed in an auxiliary manner.
> For example, Lei et al. [25] leverage LLMs to describe the status of human body parts given an HOI label, and utilize VLMs to enhance interaction recognition.
> To verify the effectiveness of VLMs and LLMs, we conducted two additional experiments.
> First, we employ CLIP [R3] to learn specific body-parts using text embeddings derived from body-part names.
> To this end, text prompts are constructed in the form of "A photo of a person [body part]".
> As summarized in Table R1, our method consistently outperformed this CLIP-guided variant, particularly in mR@25 and average recall (AR), suggesting that CLIP guidance is less effective in providing fine-grained spatial cues than pose estimators.
> We attribute this performance gap to the relatively weak spatial reasoning capabilities of current VLMs, which are primarily trained via the image-text contrastive learning.
> Second, we evaluated LLaVA-1.6-vicuna-7B [R4], a recent MLLMs, on the NVI test set.
> To support LLaVA, we provided ground-truth group bounding boxes and cropped the image accordingly before querying the model to identify interactions.
> Despite this favorable setup, LLaVA achieved only 31.88 AR, which is far lower than our method (Table R2).
> We believe this is because MLLMs are not trained specifically for fine-grained spatial reasoning and may not possess the ability to model group-level interactions.
> Nevertheless, we agree that the zero-shot capability of such models offer promising directions for future research.
>
> **Table R1: Comparison with the pose supervision and VLM supervision.**
> | Setting               | mR@25 | mR@50 | mR@100 | AR    |
> |-----------------------|-------|-------|--------|-------|
> | **Pose supervision (Ours)** | **59.43** | 76.62 | **87.43** | **74.49** |
> | VLM supervision        | 55.18 | **76.94** | 86.96 | 73.02 |
>
> **Table R2: Comparison with LLMs.**
> | Method | mR@25 | mR@50 | mR@100 | AR    |
> |--------|-------|-------|--------|-------|
> | **Ours**   | **63.59** | **80.62** | **91.34**  | **78.52** |
> | LLaVA  |  13.44 | 28.98 | 53.21  |  31.88 |
>
>
> ## Can this method be applied to other datasets and downstream tasks? (+stronger baselines)
> We thank the reviewer for the constructive suggestions.
> We initially considered GP-Static [R5] as suggested, but found that it is not publicly available.
> As an alternative, we have conducted experiments on Caf\'e [22], a more recent and challenging benchmark for group activity detection that emphasizes multi-group scenarios, similar to JRDB-Act [10].
> As shown in Table R3, our model outperforms stronger baselines such as Caf\'e-base [22], JRDB-base [10], and HGC [39] in terms of both Group mAP and Outlier mIoU.
> These results demonstrate the effectiveness of our method in the related task and further suggest that part-aware representations and bottom-up group reasoning not only benefit fine-grained social interaction detection, but also contribute to group activity understanding tasks.
>
> **Table R3: Comparison with the previous methods on Café detection-based setting.**
> | Method        | Group mAP$_{1.0}$ (view) | Group mAP$_{0.5}$ (view) | Outlier mIoU (view) | Group mAP$_{1.0}$ (place) | Group mAP$_{0.5}$ (place) | Outlier mIoU (place) |
> |---------------|----------------------|------------------------|----------------------|------------------------|------------------------|------------------------|
> | Joint [9]    | 9.14                 | 31.83                  | 42.93                | 6.08                   | 18.43                  | 2.83                   |
> | JRDB-base [10] | 12.63                | 35.53                  | 31.85                | 8.15                   | 22.68                  | 33.03                  |
> | HGC [39]       | 6.77                 | 31.08                  | 57.65                | 4.27                   | 24.97                  | 57.70                  |
> | Cafe-base [22] | 14.36              | 37.52                | 63.70              | 8.29                 | 28.72                | 59.60                |
> | **Ours**      | **18.23**            | **46.88**              | **67.62**            | **10.65**              | **39.03**              | **63.60**              |
>
> ## Analysis on the impact of data imbalance
> > One of the problems with the group activity datasets is the data imbalance problem. It would be good to see the results over different classes to see if the method is performing equally well and where are the limitations.
>
> We appreciate the reviewer’s suggestion to further analyze the impact of data imbalance.
> Table R4 compares our method and NVI-DEHR [44] in a class-wise manner.
> Among the 22 interaction classes, the two most underrepresented categories, *beckon* and *palmout*, exhibit notably low performance for both methods due to their rarity.
> Nonetheless, our method achieves notably higher recall (66.67\% and 22.22\%) compared to NVI-DEHR (16.67\% and 11.11\%), suggesting improved robustness to data imbalance.
> We attribute this improvement to the use of part-aware representations, which allow the model to explicitly focus on the specific body parts. Unlike holistic representations that may overlook infrequent combinations of body parts, part-aware modeling enables better generalization to individual body parts and consequently leads to more reliable detection of fine-grained interactions such as *beckon* and *palmout*, even under limited training examples for these classes.
>
> **Table R4: Class-wise recall (%) comparison between Ours and NVI-DEHR.**
> | Method     | neutral | anger | smile | surprise | sadness | fear  | disgust | wave  | point | beckon | palmout |
> |------------|---------|-------|--------|----------|---------|-------|---------|--------|--------|--------|---------|
> | **Ours**     | **93.17** | 72.38 | 92.29  | 82.35    | 76.06   | 66.92 | **81.24** | **72.22** | **81.59** | **66.67** | **22.22** |
> | NVI-DEHR  | 92.75  | **72.83** | **94.57**  | **82.52**  | **78.35** | **74.42** | 70.49  | 64.14  | 73.18  | 16.67  | 11.11  |
> | Method     | arm-crossing | leg-crossing | slouching | arms-akimbo | bowing | gaze-aversion | mutual-gaze | gaze-following | hug   | handshake | hit      |
> | **Ours**     | **88.19**     | **87.40**      | **88.03**   | **87.79**     | **88.81** | **70.56**       | **80.67**     | 78.91           | 73.74 | **74.68**   | **100.00** |
> | NVI-DEHR  | 79.79       | 85.91        | 55.13     | 70.34       | 84.28  | 60.89         | 78.34       | **83.20**       | **83.73** | 74.28     | **100.00** |
>
> ## Unclear output of the method
> > It is not clear, until the experimental results, the output of the method. In other words, what is the goal of social reasoning? However, if the problem definition can be introduced clearly in the beginning, it would be helpful for the readers.
>
> We thank the reviewer for this helpful comment.
> To improve clarity, we will include a figure in the introduction section that illustrates the target task and its expected output.
> In addition, we will revise the description in Line 31–39 and Section 3.1 to make the task setup and the structure of the output more transparent and accessible to future readers.
>
> ---
> [R1] Lopez-Paz et al., Unifying distillation and privileged information, ICLR 2016.
>
> [R2] Wan et al., Pose-aware Multi-level Feature Network for Human Object Interaction Detection, ICCV 2019.
>
> [R3] Radford et al., Learning Transferable Visual Models from Natural Language Supervision, ICML 2021.
>
> [R4] Liu et al., Visual Instruction Tuning, NeurIPS 2023.
>
> [R5] Chang et al., Gaze pattern recognition in dyadic communication, ETRA 2023.

---

> > ### Comment · Reviewer_tcQL · 2025-08-01
> > **Thanks authors for the response**
> >
> > The reviewer thanks the authors for their hard work and their detailed response.
> >
> > While the reviewer acknowledges that this paper addresses an important topic in advancing our understanding of automatic social reasoning, they believe it is not a good fit for this venue, which primarily emphasizes contributions to machine learning methodology. Although the use of pose-guided approaches is relevant, it is not a novel concept and has been extensively explored in other contexts. Its application to this particular problem, while interesting, does not represent a sufficiently innovative contribution for NeurIPS.
> >
> > Additionally, the paper's evaluation is limited to a single dataset with a narrow range of social interaction classes, all within the image domain. This further constrains its generalizability and impact.
> >
> > Overall, the paper is technically sound and well-written. However, given the considerations above, it does not meet the bar for acceptance at NeurIPS in the reviewer's opinion. Therefore, the reviewer maintains a borderline reject recommendation.

---

> > > ### Author Response · Authors · 2025-08-02
> > >
> > > We thank the reviewer for the timely response, and recognizing our work addresses an important topic.
> > >
> > > Regarding the concern about evaluation on a single dataset, we would like to draw the reviewer’s attention to **Table R3** in our rebuttal, where we report experimental results on the Café dataset [22]—a large-scale video benchmark for multiple group activity understanding.
> > > This experiment demonstrates that our method is effective not only for fine-grained social interaction detection in still images (NVI-DET), but also for group activity understanding in videos, highlighting its broader applicability to social reasoning benchmarks.
> > >
> > > If there are any remaining concerns not sufficiently addressed in our rebuttal, we would be happy to clarify further.

---

> > > > ### Comment · Reviewer_tcQL · 2025-08-02
> > > > **thanks**
> > > >
> > > > Many thanks for pointing out. However, I found this rather confusing. because it is not clear to me now how your method can be extended to handle videos now. it seems this paper requires major work to address the comments adequately.

---

> > > > > ### Author Response · Authors · 2025-08-03
> > > > >
> > > > > We thank the reviewer for the follow-up comment, and we apologize for any confusion caused by our additional experiment on the Café dataset.
> > > > >
> > > > > To clarify, our method was not modified to perform temporal modeling; rather, we applied it as-is, in a frame-wise manner.  The purpose of the Café experiment was to address the reviewer’s earlier concern regarding evaluation on only a single dataset:
> > > > >
> > > > > > The proposed method is tested on one dataset only, which is substandard. The reviewer understands that the datasets for this problem might be limited. However, the authors could test this on gaze communication datasets such as GPStatic. Alternatively, there is a dataset for group activity recognition called JRDB dataset, https://jrdb.erc.monash.edu.
> > > > >
> > > > > Specifically, we aimed to demonstrate that our method, although developed for fine-grained social interaction detection, is also applicable to coarse-grained group activity understanding.
> > > > > Despite the lack of temporal modeling, our method outperformed prior methods on Café, many of which are specifically designed to handle video inputs.
> > > > > We believe this result highlights the effectiveness and generalizability of our part-aware reasoning, and its potential value for broader social reasoning tasks.
> > > > >
> > > > > We hope this clarifies the interpretation of the Café experiment, and we are happy to elaborate further if needed.

---

### Note · Authors · 2025-08-13

We sincerely thank the ACs and reviewers for their time, constructive and insightful comments.
Our work was recognized for addressing an important and growing topic in fine-grained social interaction detection (tcQL, UmiK), introducing a bottom-up modeling (iWA3) and part-aware feature enhancement via pose guidance (all), achieving strong benchmark performance (UmiK), and being technically sound and well written (tcQL, UmiK).
We summarize below the key points from the discussions and our final clarifications.

## Novelty (tcQL, UmiK)

The primary novelty of our method lies in **learning part-aware representations using human pose as privileged information**—specifically for fine-grained social interaction detection.
Our method softly guides the model’s attention to the expected regions of each body part rather than enforcing exact localization, effectively capturing subtle cues and also robust to pose estimation errors.
This approach is the first of its kind for fine-grained social interaction detection. Moreover, to our knowledge, neither group activity understanding nor HOI-DET, which are closely related tasks, has employed human pose in this manner so far.
This unique approach leads to technical differences from previous work using human pose [R2, 46], and allows to improve performance without any additional computational burden in testing.

##  Applicability to other tasks (tcQL, iWA3)

To demonstrate that our method is not limited to a single image-based dataset, we extended our evaluation to the Café benchmark for group activity understanding (**Table R3**).
Although our method was not modified to perform temporal modeling, it outperformed prior methods including those with temporal modeling, showing that our part-aware reasoning generalizes beyond fine-grained social interaction detection to group activity understanding.

## Comparisons with MLLMs (tcQL, UmiK)

In response to reviewers’ suggestions, we evaluated MLLMs on the NVI dataset; to be specific, we tested a pretrained LLaVA-1.6 as-is or fine-tuned it for the task (**Table R7**).
Even under a favorable setup where ground-truth boxes are provided, LLaVA achieved far lower than our method—suggesting that a naïve adaptation of MLLMs is insufficient for this task.
Our method explicitly models individuals, their part-level representations, and their relations, which are particularly effective for distinguishing subtle social cues that current general-purpose MLLMs struggle to capture.

---

### Decision · Program_Chairs · 2025-09-17

**Decision:**

Accept (poster)

**Comment:**

This paper presents a part-aware bottom-up group reasoning framework that leverages body cues and social signals to more accurately detect fine-grained social interactions, achieving state-of-the-art results on the NVI dataset.

Summary of Strengths:
- The work tackles the important problem of fine-grained social interaction detection and has real-world applications.
- The proposed bottom-up reasoning approach, progressing from individuals to pairwise interactions to groups offers a natural way to infer group structures,
- The proposed approach achieves state-of-the-art results on the NVI dataset.
- The paper is well-written and easy to follow.

Summary of Weaknesses:
- The work mainly follows standard pipelines and has incremental novelty.
- Experimentation includes only one dataset and hence it is not clear whether the approach will generalize.
- The baselines could be stronger.

Overall, the paper proposes an effective method for modeling social interactions and will likely be useful to the community. Its incremental novelty and limited evaluation (dataset and baselines) were concerning to the reviewers, but authors added clarifications and new results during the rebuttals, that were found useful by the reviewers.